# IRE1 RNase controls CD95-mediated cell death

Diana Pelizzari-Raymundo [1,2], Victoria Maltret[1,2,3], Manon Nivet [1,2,3], Raphael Pineau[1,2], Alexandra Papaioannou [1,2], Xingchen Zhou [1,2], Flavie Caradec[1,2], Sophie Martin [1,2], Matthieu Le Gallo [1,2], Tony Avril [1,2], Eric Chevet [1,2] & Elodie Lafont [1,2 ✉]

## Abstract

**Signalling by the Unfolded Protein Response (UPR) or by the Death Receptors (DR) are frequently activated towards pro-tumoral outputs in cancer. Herein, we demonstrate that the UPR sensor IRE1 controls the expression of the DR CD95/Fas, and its cell death-inducing ability. Both genetic and pharmacologic blunting of IRE1 activity increased CD95 expression and exacerbated CD95L-induced cell death in glioblastoma (GB) and Triple-Negative Breast Cancer (TNBC) cell lines. In accordance, CD95 mRNA was identified as a target of Regulated IRE1-Dependent Decay of RNA (RIDD). Whilst CD95 expression is elevated in TNBC and GB human tumours exhibiting low RIDD activity, it is surprisingly lower in XBP1s-low human tumour samples. We show that IRE1 RNase inhibition limited CD95 expression and reduced CD95-mediated hepatic toxicity in mice. In addition, overexpression of XBP1s increased CD95 expression and sensitized GB and TNBC cells to CD95L-induced cell death. Overall, these results demonstrate the tight IRE1-mediated control of CD95-dependent cell death in a dual manner through both RIDD and XBP1s, and they identify a novel link between IRE1 and CD95 signalling.**

**Keywords** CD95; Cell Death; ER Stress; Unfolded Protein Response; IRE1
**Subject Categories** Autophagy & Cell Death; Cancer; RNA Biology

## Introduction

Throughout tumour development, cancer cells are subjected to both intrinsic and extrinsic stresses with which they must cope to survive and proliferate. Adapted tumour cells survive this selection pressure through the activation of specific signalling pathways. Amongst those are the signalling engaged by the death receptors (DR) and the unfolded protein response (UPR), an adaptive response to endoplasmic reticulum (ER) stress. The DR TRAIL-R1/2 and CD95 (also known as FAS) contribute to the immunosurveillance towards cancer or infected cells by their ability to induce cell death upon engagement by their ligand, TRAIL and CD95L, respectively (Von Karstedt et al, 2017; Rossin et al, 2019;

Risso et al, 2022). In some tumour cells however, engagement of CD95 or TRAIL-R1/2 fails to induce cell death and promotes pro-tumorigenic cellular outcomes. Similarly, constitutive UPR activation is often observed in cancer cells, resulting from adaptation to various stresses including oncogenic insults, aneuploidy or nutrient deprivation. This constitutive activation of the UPR does not result in cell death, as would be the case in normal cells, but rather allows cancer cells to thrive through activation of pro-tumoral outcomes (Almanza et al, 2019; McGrath et al, 2021). Thus, both UPR and DR signalling contribute to tumour progression and relapse in various cancer models including glioblastoma (GB) (Doultsinos et al, 2019; Drachsler et al, 2016; Kleber et al, 2008; Le Reste et al, 2020; Lhomond et al, 2018; Quijano-Rubio et al, 2022; Obacz et al, 2023) and Triple Negative Breast Cancer (TNBC) (Chen et al, 2014; Fritsche et al, 2015; Harnoss et al, 2020; Logue et al, 2018). The UPR consists of three core signalling branches initiated by the activation of three sensors upon ER stress. These sensors are ATF6α (for Activating Transcription Factor 6 alpha), IRE1α (for Inositol-Requiring Enzyme 1 alpha, referred to as IRE1 hereafter) and PERK (for Protein kinase R-like ER Kinase). IRE1 transduces its signalling through at least a dual enzymatic process including IRE1 kinase and IRE1 RNase. IRE1 RNase activity drives (i) the unconventional splicing of *XBP1* mRNA together with the tRNA ligase RtcB, ultimately leading to the expression of the transcription factor XBP1s, which in turn promotes the expression of multiple genes aimed at restoring ER homeostasis and (ii) the degradation of RNA through Regulated IRE1-Dependent Decay of RNA (RIDD). RIDD leads to both cytotoxic and non-cytotoxic cellular outcomes (Maurel et al, 2014). In addition to their individual roles in controlling cell fate, both UPR and DR signalling pathways are functionally intertwined (Lafont, 2020; Stöhr et al, 2020a). Indeed, TRAIL-R2, in some cases TRAIL-R1 and TRAIL, are upregulated downstream of the PERK/ATF4 and/or CHOP axis, in various ER stress conditions and accordingly TRAIL-R signalling can participate to cell death induced upon ER stress (Cazanave et al, 2011; He et al, 2002; Iurlaro and Munoz-Pinedo, 2016; Iurlaro et al, 2017; Jiang et al, 2007; Lafont, 2020; Lam et al, 2018; Li et al, 2015; Lu et al, 2014; Stöhr et al, 2020b; Yamaguchi and Wang, 2004). On the contrary, RIDD limits TRAIL-R2-induced signalling by reducing the abundance of its mRNA (Lu et al, 2014). Moreover, TRAIL-R2 was recently shown to be directly activated intracellularly by misfolded proteins and therefore signals apoptosis from intracellular compartments (Lam et al, 2020). Although the relationships

[1]Inserm U1242, University of Rennes, Rennes, France. [2]Centre de Lutte Contre le Cancer Eugène Marquis, Rennes, France. [3]These authors contributed equally: Victoria Maltret, Manon Nivet. ✉E-mail: elodie.lafont@inserm.fr

between ER stress and TRAIL-R have been relatively well explored, whether and how IRE1 and CD95 signalling are also linked remains however, unclear. We have recently identified CD95 mRNA as being cleaved by IRE1 RNase in an in vitro RNA cleavage assay (Lhomond et al, 2018). Herein, we investigate the impact of IRE1 RNase activity in CD95 signalling, revealing a previously unrecognised dual functional link between these pathways.

# Results

## Basal and ER-stress induced IRE1 activation limits CD95 expression in GB and TNBC cells

We have recently identified CD95 mRNA as being cleaved by IRE1 in an in vitro RNA cleavage assay (Lhomond et al, 2018). To evaluate if CD95 mRNA expression levels depend on IRE1 activity in a cellular context, we used the U87 GB cell line, which displays a constitutive activation of IRE1 (Lhomond et al, 2018). Expression, in these cells, of a dominant-negative (DN) form of IRE1, which represses the activation of endogenous IRE1, and thus its RNase activity (Nguyên et al, 2004; Pluquet et al, 2013), led to an increased basal level of CD95 mRNA (Fig. 1A). Accordingly, cell surface expression of CD95 was heightened in IRE1 DN-overexpressing U87 cells as compared to WT cells (Fig. 1B). Similar results were obtained in two primary GB cell lines, RADH87 and RADH85 (Avril et al, 2012) in which expression of IRE1Q780*, a mutant of IRE1 devoid of both kinase and RNase domains that blunts IRE1 signalling, led to an increased expression of CD95 mRNA (Fig. 1A) as well as of total and cell surface (Figs. EV1A and 1B) CD95 protein. To determine if exogenous activation of IRE1 impacts on CD95 expression, we next analysed CD95 mRNA expression levels in U87 cells and in the TNBC cell line SUM159 treated or not with ER stress inducers. In this context, both MG-132 and tunicamycin (TM) prompted CD95 mRNA degradation, which was prevented by treatment with MKC-8866, a pharmacological inhibitor of IRE1 RNase activity (Volkmann et al, 2011) (Fig EV1B,C). This effect was also confirmed using a second IRE1 inhibitor, Z4, which targets the kinase activity of IRE1 and thus inhibits its activity through a mechanism completely different to that of MKC-8866 (Pelizzari-Raymundo et al, 2023), and efficiently repressed the RIDD activity of IRE1 towards CD95 (Fig. EV1D). A consistent result was obtained regarding CD95 protein expression. Indeed, ER stress induced by TM or thapsigargin (TG) provoked a decrease in CD95 protein levels (Fig. 1C–F) which was reverted by a treatment with MKC-8866. Of note, and in accordance with previous literature, tunicamycin treatment was also accompanied by a decrease of N-glycosylated CD95 (Shatnyeva et al, 2011). Overall, these results highlight that both constitutive and ER stress-induced IRE1 RNase activities limit CD95 expression in the tested cellular models.

## IRE1 cleaves CD95 mRNA in vitro

To test whether CD95 mRNA is a target of RIDD, we first evaluated the ability of recombinant IRE1 to cleave CD95 mRNA in vitro. This revealed that CD95 mRNA is indeed directly targeted by IRE1 RNase activity (Fig. 2A). We then searched the sequence of CD95

mRNA for the presence of potential RIDD cleavage sites. Such sites have been reported to both (i) reside in hairpin loop structures and (ii) display a consensus sequence CNG/CAGN (Moore and Hollien, 2015; Oikawa et al, 2010; Yoshida et al, 2001). However, no such site was identified in the CD95 mRNA sequence. We, therefore, expanded our search to additional potential sequences beyond this classical consensus. These degenerate sequences (CAACAA, CAGCUC, CUGCAU and CUGGCG) can be targeted by IRE1 in vitro when displayed on hairpin loops as recently identified in our laboratory (Voutetakis, In preparation). Based on this second analysis, we identified two potential cleavage sites, one located within the CD95 ORF and the other in the 3'UTR of CD95 mRNA (Fig. 2B). Noteworthy, PCR amplification of 136 bp encompassing the CAACAA was not significantly affected when performed using RNA incubated with recombinant IRE1 as template, whilst that of a 121 bp region comprising the CUGCAU site was affected (Fig. 2C,D). This phenomenon was not observed for several additional RNAs (Appendix Fig. S1), implying that the effect observed on CD95 mRNA follows a specific trend which cannot be considered as a common one. Overall, these data therefore argue that the CUGCAU site might indeed be targeted by IRE1 in vitro. Taken together, these results indicate that CD95 mRNA is a bona fide substrate of IRE1 RNase in vitro.

## CD95 is not a general determinant of ER stress-induced cell death

Since the expression of another DR, TRAIL-R2, is also regulated by IRE1 (Lu et al, 2014) and that TRAIL-R2 signalling impacts on ER stress-induced cell death, we next hypothesized that CD95 may also influence the sensitivity of cells to ER stress. To test this hypothesis, we used U87 cells in which CD95 was knocked-down using RNA interference as well as MDA-MB-231 TNBC cells WT or KO for CD95 (Appendix Fig. S2A,B). Whereas CD95 depletion led to a reduced sensitivity to TM-induced loss of viability in both cell lines, it did not impact any of these cells' ability to die in response to TG, MG-132 and Brefeldin A (Figs. 3A, EV2A and Appendix Fig. S2A,B). These results, therefore, imply that while CD95 might contribute to cell death in response to some specific ER homeostasis insults or to aberrant protein N-linked glycosylation, this DR is not a universal determinant of cells' sensitivity to ER stress-induced cytotoxicity.

## IRE1 RNase activity limits CD95L-induced caspase-activation and cell death

Since our cell-based results indicate that RIDD limits CD95 expression, we hypothesized that IRE1-mediated depletion of CD95 would repress CD95L-induced cell death. Indeed, U87 cells expressing IRE1 DN were markedly sensitized to CD95L-induced loss of viability (Appendix Fig. S3A), which was indeed due to increased cell death (Fig. 3B). In line with this observation, activation of the initiator caspase-8 and the effector caspase-3, along cleavage of the latter's substrate PARP-1 happened earlier and were more pronounced in IRE1 DN-overexpressing U87 cells as compared to WT cells (Fig. 3C), further pointing out that IRE1 controls an early event in CD95 cell death signalling. Noteworthy, RADH85 cells expressing IRE1Q780* were also sensitized to CD95L-induced loss-of-viability (Appendix Fig. S3B), displaying

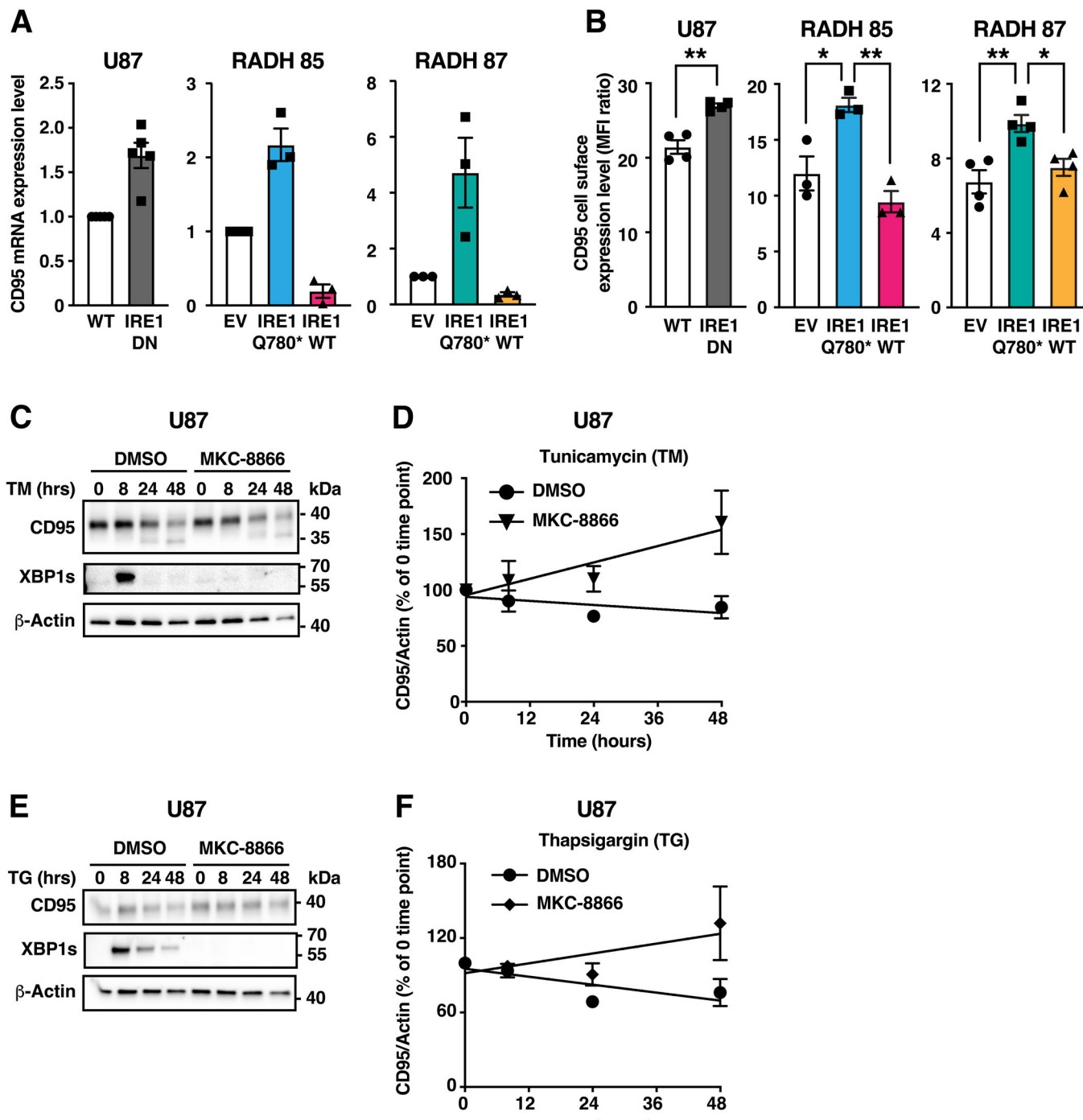

**Figure 1. IRE1 regulates CD95 expression in TNBC and GB cells.**

(**A**) mRNA was extracted from WT or IRE1 DN-expressing U87 cells and from empty vector (EV), IRE1WT- or IRE1Q780*-expressing RADH85 and RADH87 cells. CD95 mRNA was quantified by RT-qPCR and normalized to GAPDH. Mean ± SEM, $n = 3$–5. (**B**) CD95 cell surface level expression was evaluated by flow cytometry. Mean of MFI ratio ± SEM, $n = 3$–4. Unpaired $t$-test for U87 (**$p = 0.001126$); one-way ANOVA with Tukey multiple comparison correction for RADH85 (*$p = 0.018416$ and **$p = 0.003549$) and RADH87 (*$p = 0.0263$ and **$p = 0.0054$). (**C,D**) U87 cells pre-treated for 2 h with MKC-8866 (30 µM) as indicated were further treated with 500 ng/mL tunicamycin for the indicated times. Lysates were analysed using western blot. (**C**) One representative experiment out of three independent ones is shown. (**D**) Quantification for three independent experiments is depicted. Mean ± SEM. (**E,F**) U87 cells pre-treated for 2 h with MKC-8866 (30 µM) as indicated were further treated with 50 nM thapsigargin for the indicated times. Lysates were analysed using western blot. (**E**) One representative experiment out of three independent ones is shown. (**F**) Quantification for three independent experiments is depicted. Mean ± SEM. Source data are available online for this figure.

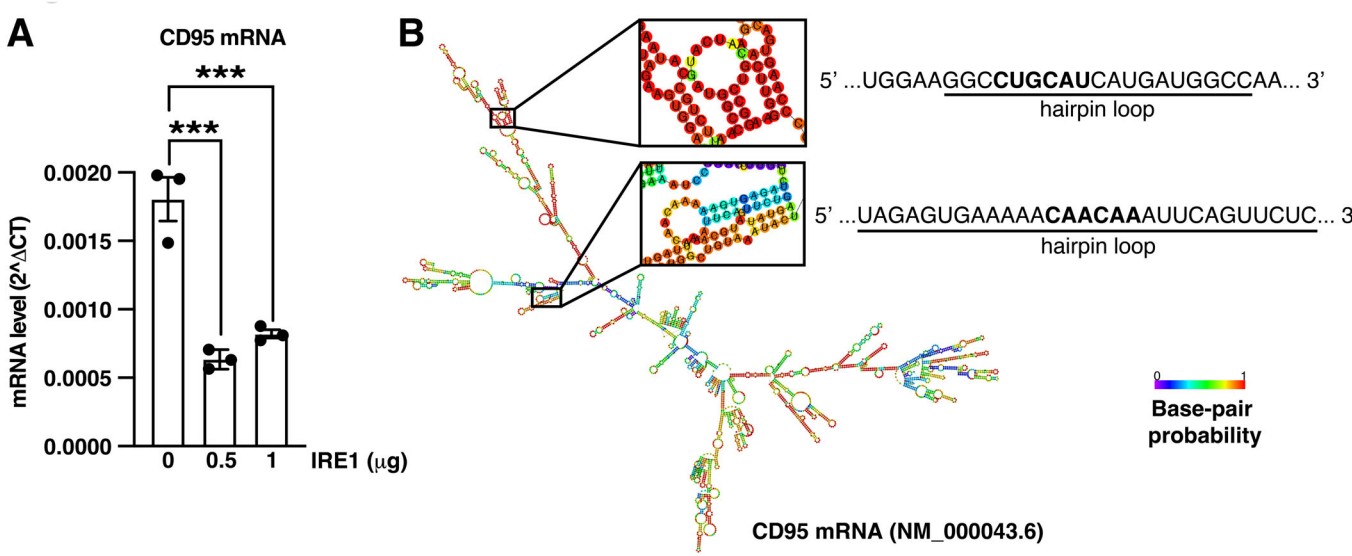

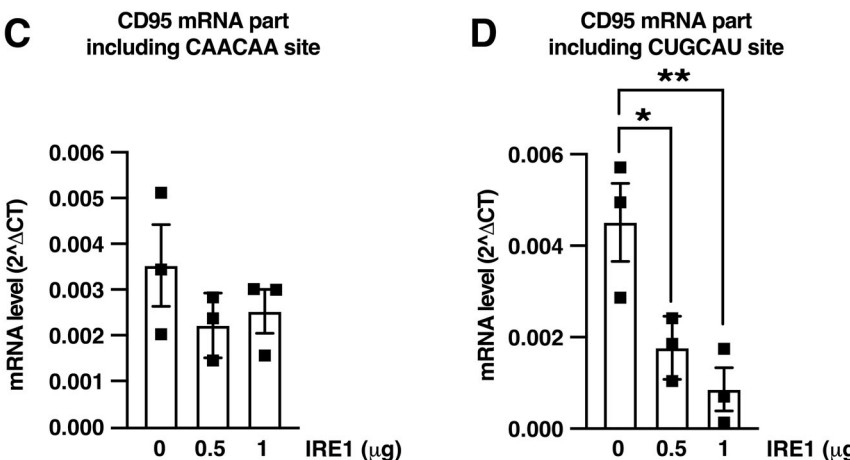

**Figure 2. IRE1 cleaves CD95 mRNA in vitro.**

(A) RNA (2 µg) extracted from U87 cells was incubated with the indicated amounts of recombinant IRE1 for 1 h. CD95 mRNA was then quantified by RT-qPCR and normalized to GAPDH. Mean ± SEM, n = 3. One-way ANOVA with Dunnett multiple comparison correction, ***p = 0.0003 (0 vs 0.5 µg IRE1 groups), ***p = 0.0007 (0 vs 1 µg IRE1 groups) (B) Predicted folded structure of CD95 mRNA. The two predicted cleavage sites within hairpin loops are highlighted. (C,D) RNA (2 µg) extracted from U87 cells was incubated with the indicated amounts of recombinant IRE1 for 1 h. 136-bp (C) and 121-bp (D) parts of CD95 mRNA including the indicated potential cleavage sites were then quantified by RT-qPCR and normalized to GAPDH. Mean ± SEM, n = 3. One-way ANOVA with Dunnett multiple comparison correction, (D) *p = 0.0331, **p = 0.0096. Source data are available online for this figure.

heightened cell death (Fig. 3D). Binding of CD95L to its receptor CD95 triggers the formation of a CD95-associated complex called the Death Inducing Signalling Complex (DISC) (Kischkel et al, 1995) required for cell death induction and comprising the adaptor FADD, caspase-8 and cFLIP. RADH85 cells expressing IRE1Q780* displayed increased DISC formation upon CD95L treatment compared to both EV- and IRE1 WT- expressing cells (Fig. 3E), advocating for an early role of IRE1 in control of CD95 signalling. Despite increased expression of CD95 (Fig. 1B), RADH87 expressing IRE1Q780* were not sensitized to CD95L-induced loss of viability (Appendix Fig. S3C), leading us to postulate that an additional cell death-inhibitory checkpoint, downstream of CD95,

might be active in these cells. To test this hypothesis, we targeted the anti-apoptotic proteins cIAP1/2 and XIAP, which are known negative regulators of CD95-mediated cell death (Geserick et al, 2009; Jost et al, 2009). To this end, we utilized the SMAC mimetic Birinapant, which potently inhibits cIAP1 and, albeit to a lesser extent, cIAP2 and XIAP (Benetatos et al, 2014; Condon et al, 2014). RADH87 expressing IRE1Q780* were notably sensitized to CD95L-induced loss of viability (Appendix Fig. S3C) and increased cell death (Fig. EV2B) when co-incubated with Birinapant as compared to control or IRE1 WT-overexpressing cells. Taken together, these data argue for an early role of IRE1 in limiting CD95L-induced cell death signalling.

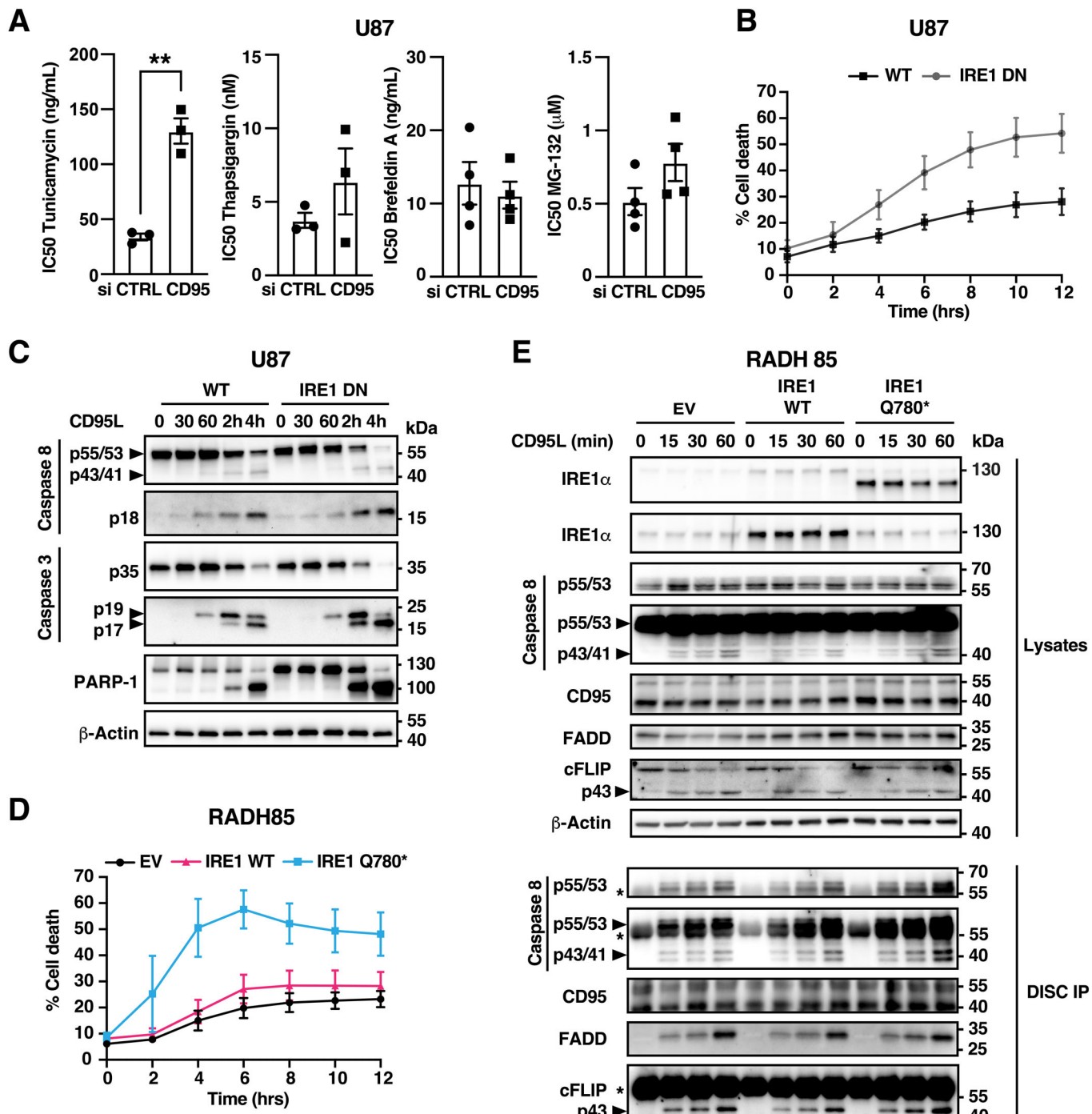

**Figure 3. CD95 is not a universal determinant of ER-stress induced cell death whilst IRE1 RNase activity limits CD95L-induced cell death.**

(A) U87 were transfected with siRNA control or targeting CD95. 48 h later, cells were treated for 48 h with the indicated ER stress inducers. Viability was determined using an MTT assay. The relative IC50 calculated for each independent experiment is represented (see also Appendix Fig. S2A). Mean ± SEM, n = 3–4. **p = 0.013, unpaired t-test (B). U87 WT or expressing IRE1DN were treated with 1 µg/mL CD95L for 12 h. % of cell death was defined as the % of Cytotox red-positive cells as detected by the Incucyte. Mean ± SEM of three independent experiments. (C) U87 WT or expressing IRE1DN were treated with 250 ng/mL CD95L for the indicated times. Lysates were analysed using western blot. One experiment representative of three independent ones is shown. (D) RADH85 control (EV), stably expressing IRE1Q780* or IRE1WT were treated with 500 ng/mL CD95L for 12 h. % of cell death was defined as the % of Cytotox red-positive cells as detected by the Incucyte. Mean ± SEM of three independent experiments. (E) Empty vector (EV), IRE1WT- or IRE1Q780*-expressing RADH85 were treated with 500 ng/mL CD95L for the indicated times. The DISC was immunoprecipitated using an anti-CD95 antibody prior to western blot analysis. One experiment representative of two independent ones is shown. *Indicates an unspecific band. Source data are available online for this figure.

## Impact of IRE1 RNase inhibition in CD95-mediated cell death in an acute liver injury mouse model

To determine whether IRE1 RNase activity controls CD95 expression and signalling in vivo, we first evaluated the impact of intra-peritoneal injection of MKC-8866 on CD95 expression in mouse livers (Fig. 4A). In contrast with the results previously obtained in cancer cell lines (Figs. 1–3), this analysis indicated that the pharmacological inhibition of IRE1 RNase led to decreased CD95 protein expression in vivo (Fig. 4B). To investigate whether this unexpected reduction of CD95 expression levels coincided with a decreased sensitivity to CD95-mediated cell death, we next used the anti-CD95 agonist antibody Jo2 at a sub-lethal dose, known to induce a mild hepatitis within 6 h of treatment (Filliol et al, 2017). Mice pre-treated with MKC-8866 or vehicle were therefore injected with Jo2 or the corresponding isotype control (Fig. EV3A). First, immunohistochemical (IHC) analysis of the liver sections confirmed that inhibition of IRE1 led to a reduction in CD95

expression (Fig. EV3B). Furthermore, both HES and IHC analysis of the same liver sections using an anti-cleaved caspase-3 indicated a clear reduction of Jo2-induced liver damage in the MKC-8866-treated group as compared to vehicle-treated animals (Figs. 4C and EV3C), a result which was further confirmed by Western blot (Fig. EV3D). Taken together, these results indicate that IRE1 RNase activity promotes CD95 hepatic expression and CD95-mediated hepatotoxicity in vivo.

## Dual regulation of CD95 expression and signalling by IRE1 RNase activity

To further investigate the apparent discrepancy in the results obtained in cell lines and in vivo, we next explored how the different branches of IRE1 signalling impact on CD95 expression. Indeed, IRE1 RNase activity catalyzes both RIDD and the unconventional splicing of XBP1 mRNA, the latter resulting in the expression of the transcription factor XBP1s. We first evaluated

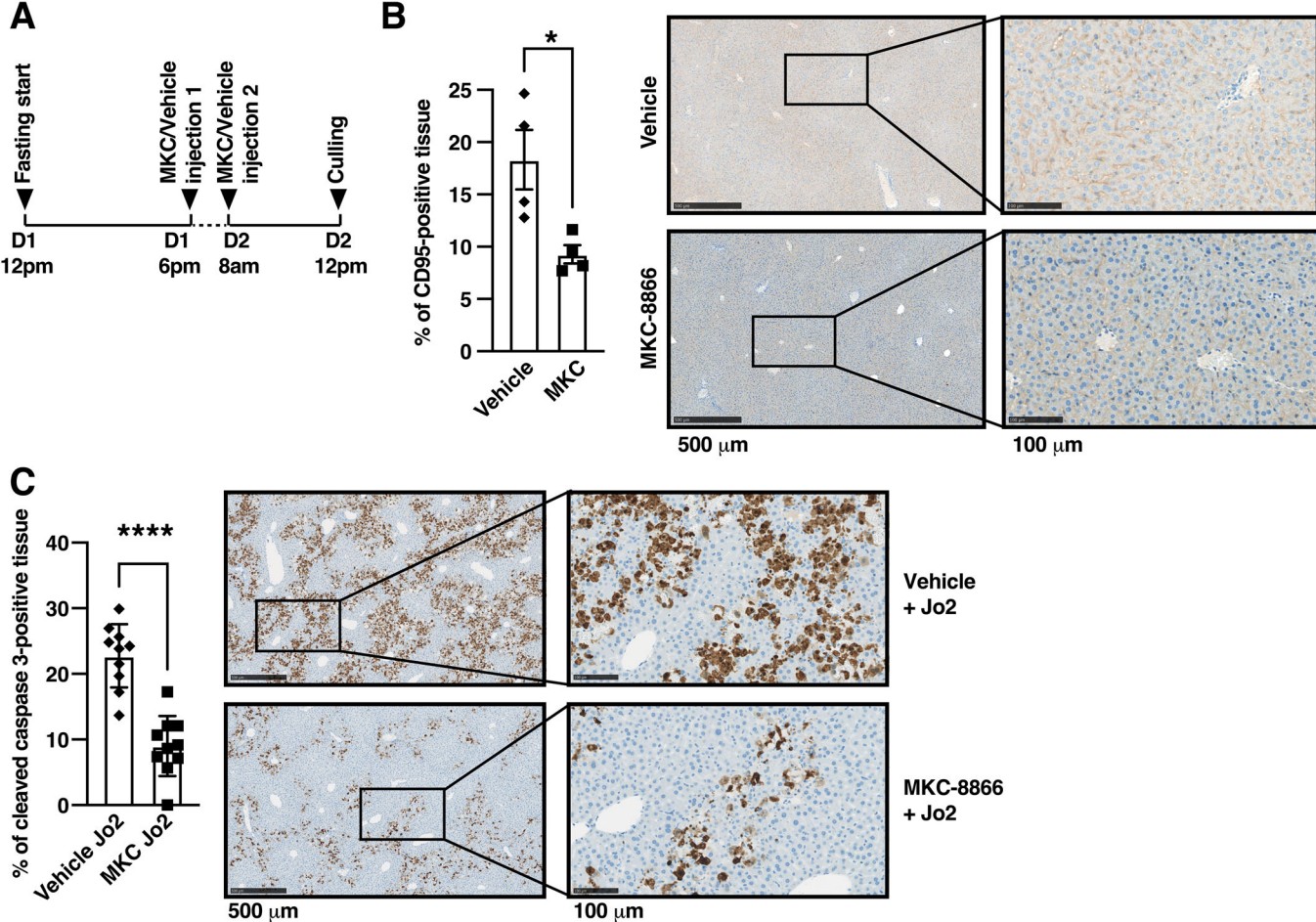

**Figure 4. IRE1 RNase inhibition limits hepatic CD95 expression and CD95-mediated cell death in mice.**

(A) Timeline of in vivo experiment 1. Eight mice were divided in two groups of 4 and repeatedly injected as indicated with either vehicle (group 1) or MKC-8866 (group 2). (B) IHC staining for CD95 on liver tissue from the two groups of mice described in A. Left: quantification of CD95 staining. Mean ± SEM of $n = 4$ mice per group; *$p = 0.0286$, with Mann–Whitney test for comparison of the two groups. Right: representative IHC image for each group. (C) IHC staining for cleaved caspase-3 on liver tissue from the two indicated groups of mice described in Fig. EV3A. Left: quantification of cleaved caspase-3 staining. Mean ± SD of $n = 10$ mice per group; five images per liver. ****$p = 0.000043$ Mann–Whitney test for comparison of the two indicated groups. Right: representative IHC image for each group. Source data are available online for this figure.

the impact of knocking-down IRE1 or XBP1 for 24 h on CD95 constitutive expression (Figs. 5A and EV4A), revealing that in these conditions, both IRE1 and XBP1 promote CD95 expression. In accord, XBP1s overexpression promoted CD95 expression (Figs. 5B and EV4B), but none of the other pro-apoptotic components

evaluated in SUM159 or U87 cells (Appendix Fig. S4). Of note, the promoter regions of both murine and human *FAS* displayed several potential XBP1s-binding sites, whilst each individual site is only partially conserved between these two species (Appendix Fig. S5 and Table S1). Together, these observations suggest that CD95

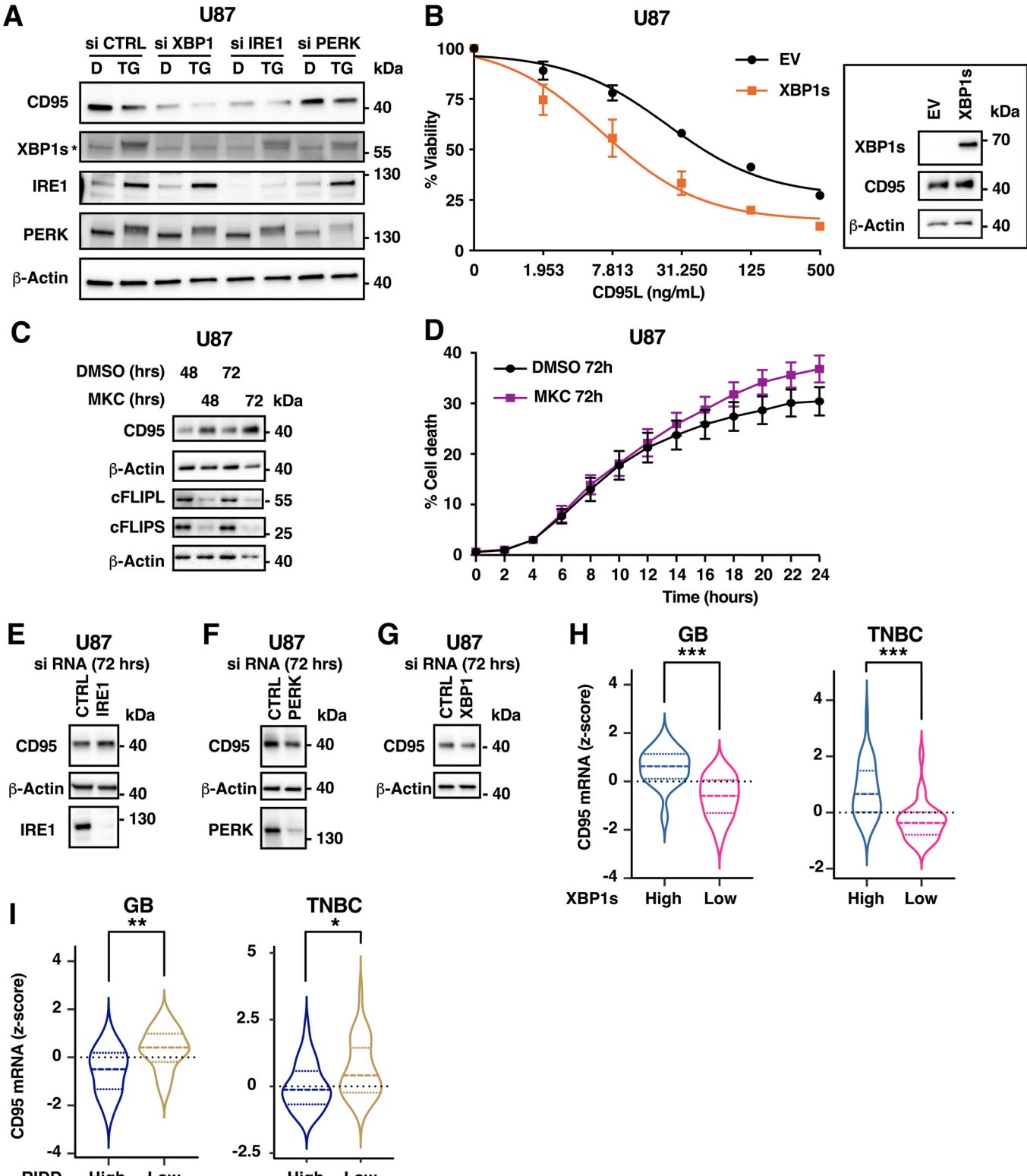

◀ **Figure 5. IRE1 dually controls CD95 expression and CD95L-induced cell death via RIDD and XBP1s and the relative activation of each branch correlates with CD95 expression in tumours.**

(A) U87 cells were transfected with siRNA control or targeting XBP1, IRE1 or PERK as indicated. 16 h later, cells were treated with DMSO (D) or 250 nM thapsigargin (TG) for 8 h. Lysates were analysed using western blot. One experiment representative of at least three independent ones is shown. (B) U87 cells were transfected with a plasmid coding for FLAG-XBP1s (XBP1s) or an empty vector (EV). 48 h later, cells were treated with the indicated concentrations of CD95L for 48 h. Viability was assessed using MTT assay and normalized to untreated cell values. Mean ± SEM of three independent experiments. Inset: western blot analyses of cell lysates 48 h post-transfection. (C) U87 cells were treated with DMSO or MKC-8866 (30 mM) for the indicated times. Lysates were analysed using western blot. One experiment representative of three independent ones is shown. (D) U87 cells treated with 30 µM MKC-8866 or DMSO were further treated with 1 µg/mL CD95L for the indicated times. Cell death was evaluated using Cytotox red positivity. Mean ± SEM, n = 3. (E–G) U87 were transfected with siRNA control or targeting XBP1, IRE1 or PERK as indicated. 72 h later, cell lysates were analysed using western blot. One experiment representative of at least three independent ones is shown. (H,I) CD95 expression z-scores of 45 GB and 62 TNBC tumours were plotted according to the RIDD activity score (H) and according to the XBP1s activity score (I). The distribution of z-score is represented as violin plots. For GB n = 45; for TNBC n = 62. Statistical difference of expression between groups was calculated using Mann–Whitney tests and the p-value is indicated (H GB ***$p = 4.3e{-}05$, TNBC ***$p = 4.1e{-}06$; I GB** $p = 0.0025$, * TNBC $p = 0.015$). Source data are available online for this figure.

could be a genuine XBP1s-target gene. Corroborating the hypothesis that XBP1s, contrary to RIDD, may enhance CD95-mediated cell death, overexpression of XBP1s in both U87 and SUM159 increased CD95L-induced cell death (Figs. 5B and EV4B). It remains to be evaluated whether this transcription factor sensitizes cells solely through upregulation of this DR, or of yet-to-be identified pro-apoptotic factors, a combination thereof or additional more indirect mechanisms.

Upon acute and intense ER stress induction, with tunicamycin or thapsigargin, CD95 expression was dramatically decreased and, in accord with the data obtained in Fig. 1C–F, this effect was limited upon knockdown of IRE1 but not of XBP1 (Figs. 5A and EV4A), which pointed to the dominance of RIDD in regulating the expression of CD95 in this context. We also evaluated whether PERK, another ER stress sensor which controls the expression of the DR TRAIL-R1/2 (Lafont, 2020), impacts on CD95 expression too. Interestingly, knockdown of PERK limited ER stress-induced decrease of CD95 protein expression without majorly affecting IRE1 levels (Figs. 5A and EV4A). We hypothesize that PERK may either promote IRE1 RIDD activity -and thus favour acute ER stress-induced decrease of CD95 expression- by a yet undefined mechanism or alternatively (and probably more likely), silencing of PERK may counteract ER stress-induced decrease of CD95 by preventing ER-stress induced protein synthesis attenuation.

Contrary to the observations made in vivo with repeated short-term treatment with MKC-8866 (Fig. 4) or upon short-term knockdown of IRE1 or XBP1 (Figs. 5A and EV4A), a long-term treatment (from 48 h onwards) of U87 cells with either MKC-8866 or the IRE1 kinase inhibitor Z4 (Pelizzari-Raymundo et al, 2023) induced an accumulation of CD95 (Figs. 5C and EV4C). In addition, long-term treatment with MKC-8866 also sensitized U87 cells to CD95L-induced cell death (Fig. 5D) and decreased the level of cFLIPL/S which may also contribute to death sensitization (Fig. 5C). Similarly, long-term knock-down (72 h) of IRE1 promoted CD95 expression, whilst both PERK and XBP1 knock-down rather repressed CD95 expression (Fig. 5E–G). In accord, long-term knockdown of IRE1 promoted CD95L-induced cell death (Fig. EV5D). We therefore deem likely that the results obtained in Fig. 1A,B, using cells overexpressing WT or dominant negative forms of IRE1 may represent models of chronic (long-term) stress. Together with the in vivo data (Fig. 4), our results point towards a role for XBP1s in promoting CD95-mediated cell death in short-term mild stress conditions (death and selection of the fittest cells). However, upon particularly short-term intense or prolonged/chronic stress conditions, the RIDD branch likely becomes predominant (survival of adapted cells). Collectively our results thus indicate a dual regulation of CD95 expression and signalling by IRE1 RNase activity.

## Dual correlation of CD95 expression by IRE1 RNase activity in human tumours

Next, to further document the possible relevance of such mechanism to pathophysiological conditions, we explored whether the activation of RIDD and XBP1s might correlate with CD95 mRNA expression levels in human tumours. Therefore, we analysed the expression of this mRNA in tumours classified according to their RIDD and XBP1s gene expression signature, as defined previously (Lhomond et al, 2018). This analysis revealed that XBP1s-high tumours present a significantly heightened expression of CD95, as compared to XBP1s-low tumours (Fig. 5H). Conversely, RIDD-high tumours present a significantly diminished CD95 mRNA level when compared to RIDD-low tumours (Fig. 5I), hence indicating a dual regulation of CD95 mRNA expression by IRE1 signalling. A broader analysis of the expression (mRNA) of other DR (TNFR1, TRAIL-R1, TRAIL-R2) revealed that most of these factors were also significantly differently expressed dependently on the RIDD and XBP1s signature (Fig. EV5A–F). Intriguingly, the expression of these receptors seemed to be overall preferentially correlated with the activation of XBP1s in TNBC tumours, whilst being overall negatively correlated with RIDD in GB tumours (Fig. EV5A–F). The predominance of each IRE1 branch towards CD95 expression, which appears to depend on the cellular ER stress status (both intensity and chronicity), could thus vary depending on the cancer type.

## Discussion

Both UPR and DR signalling contribute to the elimination of cancer cells whereas diversion of these pathways towards pro-tumorigenic outcomes can promote tumour progression. So far, extensive experimental evidence demonstrated that TRAILR-1/2-signalling and the UPR are functionally linked to control cell fate, mainly through PERK as reviewed in (Lafont, 2020; Iurlaro and Munoz-Pinedo, 2016; Stöhr et al, 2020a). For example, PERK promotes TRAIL-R1, TRAIL-R2 and TRAIL expression, thereby favoring ER stress-induced cell death (Cazanave et al, 2011; He et al, 2002; Iurlaro et al, 2017; Jiang et al, 2007; Lam et al, 2018; Li et al, 2015; Stöhr et al, 2020b; Yamaguchi and Wang, 2004;

Martin-Perez et al, 2012; Anderson et al, 2015; Martín-Pérez et al, 2014; Lindner et al, 2020; Lu et al, 2014). The extent to which TRAIL-R signalling contributes to ER stress-induced cell death depends on the cell type/context (Munoz-Pinedo and Lopez-Rivas, 2018; Glab et al, 2017; Hagenlocher et al, 2022; Favaro et al, 2023). Furthermore, it has recently been shown that TRAIL-R2 can directly bind to and get activated by misfolded proteins (Lam et al, 2020) to promote cell death from intracellular compartments (Hellwig et al, 2022). In addition, IRE1 does repress TRAIL-R2 expression through its RIDD activity (Lu et al, 2014), thereby limiting cell death. Of note, the TRAIL-R1 gene has also been identified as a target of XBP1s in a chromatin immunoprecipitation experiment (Chen et al, 2014), and it is thus tempting to speculate that the expression of this DR might be directly inducible by this transcription factor at least in some cell types. TRAIL-R1/2-emanating signals not only contribute to ER stress-induced cell death but also to cytokine production (Sullivan et al, 2020), which is also a feature that is regulated downstream of ER stress signalling (Obacz et al, 2023). With regards to CD95, it was previously reported that its expression can be induced by ER stress in macrophages, a phenomenon which was attributed to a calcium/calmodulin-dependent protein kinase IIγ (CaMKIIγ)-dependent pathway (Timmins et al, 2009). In addition, in INS-1E rat insulinoma cells, it was suggested that cyclopiazonic acid could enhance IL-1-mediated CD95 expression increase in an XBP1s-dependent manner (Miani et al, 2012). Therefore, the existence of a potential connection between ER homeostasis control and CD95 in cancer cells remained largely uncharted.

Herein, we observe that IRE1 RNase cleaves CD95 mRNA in vitro and IRE1 RIDD activity represses CD95 expression (at both mRNA and protein levels) as well as CD95L-induced cell death in TNBC and GB cell lines. Accordingly, CD95 mRNA expression is reduced in both GB and TNBC human tumours displaying an elevated RIDD activity. Unexpectedly, we observed that expression of CD95 mRNA is increased in XBP1s-high tumours. This observation coincided with the fact that knockdown of XBP1s represses CD95 expression whilst overexpression of XBP1s increases CD95 expression and sensitizes cells to CD95L-induced cell death and that repeated short-term pharmacological inhibition of IRE1 RNase represses both CD95 hepatocyte and the hepatotoxicity induced by a CD95-targeting agonistic antibody. Therefore, our study highlights a previously unrecognised and dual link between IRE1 activity and the expression and signalling of CD95.

The pathophysiological importance of such connection remains to be fully explored, but given the increasing variety of physiological and pathological conditions influenced by CD95 and/or IRE1 (Almanza et al, 2019; Risso et al, 2022), we deem it likely to be broadly relevant. One such context could be tumour progression. Indeed, the ability of IRE1 to dually regulate CD95-mediated cell death suggests that the preferential branch activated by this RNase in cancer cells could be one of the determinants of their response to endogenous immunosurveillance or even T-cell immunotherapy (Singh et al, 2020; Upadhyay et al, 2021). Furthermore, this study and previous work (Lu et al, 2014) suggests that the expression of additional DR could be controlled by IRE1. Thus, one could speculate that such a regulation could contribute to the anti-tumoral effect of IRE1 overexpression recently reported in immunocompetent mice (Martinez-Turtos

et al, 2022). Furthermore, in the context of our results, IRE1-mediated regulation of CD95 signalling might provide another sensing mechanism to eliminate unfit cells (subjected to mild UPR or to acute/intense UPR), that may favour the emergence (and the progression) of aggressive tumour cells.

CD95 can also mediate a variety of non-cytotoxic pro-tumoural cellular outcomes, including migration, production of pro-inflammatory cytokines, proliferation and regulation of cell differentiation state, which are also modulated by IRE1 signalling (Annibaldi and Walczak, 2020; Risso et al, 2022; Rossin et al, 2019). Therefore, it is likely that IRE1-mediated CD95 increased expression could contribute to tumour progression in cancer cells which display primary or acquired resistance to CD95L-induced cell death. Hence, to promote CD95L-induced cell death over non-apoptotic cellular outcomes, it will also be required to define tumour contexts in which additional cell death checkpoints (e.g., mediated by IAPs) should be alleviated in concert with IRE1 inhibition. Additional pathological conditions have been shown to be potentially regulated by both CD95 and IRE1. For example, IRE1 inhibition prevents fibrosis in an idiopathic pulmonary fibrosis (IPF) murine model (Auyeung et al, 2022; Thamsen et al, 2019). In this context, loss of CD95 promotes persistent fibrosis (Redente et al, 2020), thus pointing towards this non-cancer model to also evaluate the IRE1/CD95 relationships.

Our data indicate that CD95-dependent cell death signals are oppositely regulated by RIDD and XBP1s. Such a phenomenon could therefore represent a mechanism to fine-tune life and death decisions. Given that we observed that the expression of CD95 is also affected by modulating PERK expression, defining which UPR-dependent regulation of CD95 expression is dominant in physiological or pathological contexts will also be of interest. In addition, such dual mechanism downstream of IRE1 might not be such a rare occurrence, as suggested by our observation that the expression of additional DR (TRAIL-R1, TRAIL-R2 and TNFR1) correlates with the relative activation of XBP1s and/or RIDD in GB and TNBC. Thus, the existence of a dual regulation of these DR expression by IRE1 will also need to be formally tested. Noteworthy, recent work from the laboratory demonstrated that the E2 ubiquitin-activating enzyme UBE2D3, which mediates IRE1-dependent cytokine/chemokine expression, is also a dual target of IRE1 and that more than 10% of XBP1s target genes (identified by ChIP) may also be potential RIDD substrates (Obacz et al, 2023). Such a dual regulation has also been hinted at by additional reports focusing on insulin production and signalling. For example, in pancreatic beta-cells, IRE1 has also been suggested to repress insulin expression and secretion through RIDD (Han et al, 2009; Lipson et al, 2008), whilst promoting insulin production and folding via XBP1s (Hassler et al, 2015; Tsuchiya et al, 2018; Lee et al, 2011). Therefore, a dual regulation of gene expression downstream of IRE1 is likely a broader phenomenon, with functional relevance well beyond DR signalling.

Whilst in the context of CD95 and other DR signalling IRE1-dependent dual regulation may be an additional way to ensure timely cell death induction, it will be interesting to explore the functional consequences of modulating the other yet-to-be-validated dual targets of IRE1 signalling. On the IRE1 side, defining which combinations of parameters (e.g., oligomerization, interacting partners, intensity and duration of ER stress, nature of the stressor, cellular state or tissular context) control the activation

of each branch, just as deciphering the combined molecular characteristics of dual targets (e.g., DNA/RNA sequence and structure, RNA/DNA binding proteins, RNA cellular localisation or abundance) will also be key in broadening our understanding of IRE1 biology. Such a knowledge may also help to better define the specific tumour contexts in which pharmacological inhibition or activation of IRE1 may be beneficial or detrimental for the patient.

## Methods

### Cell lines, cell culture and reagents

U87, RADH85 and RADH87 (WT, DN, EV, IRE1WT or IRE1Q780*) and MDA-MB-231 WT or CD95 KO1 and 2 (initially named CD95 KO 5 and KO 9 respectively) were generated in our laboratory as described previously (Avril et al, 2012; Guégan et al, 2021; Lhomond et al, 2018; Nguyên et al, 2004; Pluquet et al, 2013). All cell lines were cultured in Dulbecco's modified Eagle's medium (DMEM) supplemented with 10% decomplemented FBS and 2 mM L-glutamine at 37 °C in a 5% $CO_2$ incubator. Modified RADH85 and 87 were cultured with 0.8 µg/mL or 1 µg/mL puromycin, respectively. All cells were regularly tested for mycoplasma absence. CD95L was produced and quantified in-house as described previously (Risso et al, 2023). Thapsigargin (SML1845), Actinomycin D (A9415) and Tunicamycin (T7765) were from Sigma-Aldrich. Brefeldin A (S7046), MKC-8866 (S8875), Birinapant (S7015), MG-132 (S2619) were from Selleckchem, Z4 was from Enamine (Z940452448).

### RT-qPCR

Total RNA was extracted from cells using Trizol (Thermo-Fisher Scientific, Thermo-Fisher Scientific, 15596026) according to the manufacturer's instructions. cDNA was synthesized from the total RNA using the Maxima Reverse Transcriptase enzyme, random hexamer primers, dNTP mix and the Ribolock RNase inhibitor (Thermo-Fisher Scientific). PCR was performed on the template cDNA using Phusion High-Fidelity DNA Polymerase and dNTP mix (Thermo-Fisher Scientific). Quantitative PCR was alternatively performed for the cDNA using the SYBR® Premix Ex Taq™ (Tli RNase H Plus) (TAKARA-Clontech) using a QuantStudio5 system (Applied Biosystems). The primer sequences used for these experiments are shown in Table 1.

### In vitro RNA cleavage assay

The predicted CD95 mRNA structure was obtained using the RNAfold Web server (Lorenz et al, 2011). Different concentrations of recombinant IRE1(Sino Biological) were incubated with 2 µg of RNA extracts from U87 cell line, 2 mM DTT (Merck, D3801), 20 mM ATP (Merck, A2383) in 20 mM Tris, 500 mM NaCl, 10% gly, pH7.4 for 1 h at 37 °C. After the incubation, the resulting samples were used to perform qRT-PCR as described above.

### Cell and tissue lysis and western blot

For cell lines: cells seeded in six-well plates were treated as indicated in the figure legends and lysed in 100–250 µL ice-cold RIPA buffer

(Tris-HCl 50 mM, ph7.4, NaCl 150 mM, Sodium Deoxycholate 0.5%, Triton 1%, SDS 0.1%) per well, including proteases (Protease inhibitor cocktail, Sigma-Aldrich, P8340) and phosphatase inhibitors (Phosphatase inhibitor cocktail 2, Sigma-Aldrich, P5726). After 10 min incubation on ice, lysates were briefly sonicated and cleared by centrifugation ($15,500 \times g$, 30 min, 4 °C). Lysates were collected and protein quantified using Pierce™ BCA protein assay (Thermo scientific). Laemmli sample buffer (BIO-RAD, 1610747) with 2-mercaptoethanol (BIO-RAD, 1610710) or DTT (Sigma, D0632) was added to the samples which were heated (95 °C, 5 min). For liver tissue: tissue pieces (300–450 mg) were flash-frozen using liquid nitrogen upon collection. Tissues were homogenized with a Precellys Lysing kit (Bertin Ref. P000911) according to the manufacturer's instructions. For western blot analyses of cell and tissue lysates, 10–50 µg protein was loaded on 10% or 12% acrylamide home-made or gradient (4–15% acrylamide) commercial (BIO-RAD, 4561086) Tris-Glycine-SDS gels along a protein ladder (PageRuler prestained protein ladder, Thermo scientific, 26617). Proteins were then transferred onto nitrocellulose membranes using the turbo transfer system (BIO-RAD). Membranes were saturated with TBS with 0.5% Tween-20 and 5% Milk for 1 h prior to overnight 4 °C incubation with the indicated primary antibody diluted 1/1000° in TBS-Tween-5%BSA or milk and 0.025% Sodium Azide. HRP-coupled secondary antibodies, used at 1/5000° in TBS-Tween-5% milk, were from Southern Biotech. Signal Fire™ ECL reagent (Cell signalling) or ECL Revelblot intense (OZYme, OZYB002-1000) was used and chemiluminescence signal was detected using G:Box Chemi XX6 imager from Syngene. For quantification, the Image J software was used (Schneider et al, 2012). The following primary antibodies were used for western blot: CD95 (Cell signalling, 4233, AB_2100359), Actin (Sigma-Aldrich, A5316, AB_476743), IRE1 (Cell signalling, 3294, AB_823545 and Santa Cruz, 390960, AB_2936473), XBP1s (Biolegend, 647502, AB_2241743), Caspase-8 (Adipogen, AG-20B-0057-C100, AB_2490271), Caspase-3 (Cell signalling, 14220, AB_2798429), Cleaved caspase-3 (Cell signalling, 9661, AB_2341188), FADD (Cell signalling, 2782, AB_2100484), PUMA (Cell signalling, 12450, AB_2797920), BAX (Cell signalling, 5023, AB_10557411), BAK (Cell signalling, 12105, AB_2716685), BID (Cell signalling, 2002, AB_10692485), Caspase-7 (Cell signalling, 12827, AB_2687912), Caspase-9 (Cell signalling, 9502, AB_2068621), HOIP (Ubiquigent, 68-0013-100), CYLD (Santa Cruz, sc-74435, AB_1122022), A20 (Santa Cruz, sc-166692, AB_2204516), cIAP1 (Bio-Techne, AF8181, AB_2259001), FLIPL and S (Enzo Life Sciences, ALX-804-961-0100, AB_2713915), FADD (BD Biosciences, 556402, AB_396409), RIPK1 (Cell Signalling, 3493, AB_2305314), PARP-1 (Cell Signalling, 9542, AB_2160739).

### Cell activation, lysis and DISC immunoprecipitation

The following protocol for DISC purification was adapted from (Hillert-Richter and Lavrik, 2021). In brief, RADH85 (EV, IRE1WT and IRE1Q780*) cells were treated for the indicated times with CD95L in 0.5% SVF DMEM. The cells were then lysed in 500 µL ice cold lysis buffer (50 mM Tris HCl, pH 7.4, 150 mM NaCl, 1%Triton X100) containing 1% protease inhibitors (Protease Inhibitor cocktail, Sigma-Aldrich, P8340) and 1% phosphatase inhibitors (Phosphatase Inhibitor cocktail 2, Sigma-Aldrich, P5726). After 30 min incubation on ice, the lysates were purified by centrifugation ($20,000 \times g$, 25 min, 4 °C) and incubated overnight at 4 °C on a

**Table 1. RT-qPCR primers.**

| | Forward 5′-3′ | Reverse 5′-3′ |
|---|---|---|
| CD95 | 5′-AATCCTGAAACAGTGGCAATAAA-3′ | 5′-TTTCGAACAAAGCCTTTAACTTG-3′ |
| GAPDH | 5′-AAGGTGAAGGTCGGAGTCAA-3′ | 5′-CATGGGTGGAATCATATTGG-3′ |
| CD95 (CTGCAT cleavage site) | 5′-GGACCCTCCTACCTCTGGTT-3′ | 5′-GAGGACAGGGCTTATGGCAG-3′ |
| CD95 (CAACAA cleavage site) | 5′-TGTCCAAGACACAGCAGAACA-3′ | 5′-CCAAGCAGTATTTACAGCCAGC-3′ |
| XBP1s | 5′-TGCTGAGTCCGCAGCAGGTG -3′ | 5′-GCTGGCAGGCTCTGGGAAAG -3′ |
| IRE1 | 5′-GCCACCCTGCAAGAGTATGT-3′ | 5′-ATGTTGAGGGAGTGGAGGT-3′ |
| β-actin | 5′-CATGGGTGGAATCATAATGG -3′ | 5′-AGCACTGTGTTGCGCTACAG -3′ |
| ATP5h | 5′-GGCCATTGCTAGTTCCCTGA-3′ | 5′-GCACAAGATTTCACCTTCTTCTCA-3′ |
| Tubulin | 5′-CTATGTGCCGCAGGTTCTCT-3′ | 5′-CCGAAGCCGATTCTCACCAT-3′ |
| VCP | 5′-CAGCTCATCTACATCCCACTTC-3′ | 5′-CAGCTCCAGAGAAGCCATTAG-3′ |
| HPRT | 5′- TGACCTTGATTTATTTTGCATACC -3′ | 5′-CGAGCAAGACGTTCAGTCCT-3′ |

rotating wheel with 4 µg/sample of anti-CD95 antibody (Enzo, ALX-805-038-C100). 2 mg per sample of pre-washed protein G-coupled magnetic beads (DynabeadsTM Protein G, Thermoscientific, 10004D) were added to each sample and incubated on a rotating wheel at 4 °C for 2 h. The beads were then washed six times with 1 mL of wash buffer (50 mM Tris HCl, pH 7.4, 150 mM NaCl, 1% Triton X100, protease and phosphatase inhibitors diluted 1:400). Elution was performed by adding 20 µL of loading buffer (2X Laemmli (Bio-RAD) with 200 mM DTT (Sigma-Aldrich, D0632)), and incubating the beads at 95 °C for 5 min. The eluates were diluted by half with ultrapure water before analysis by western blot.

## Cell death

U87 WT/DN and RADH87 EV/WT/Q* were seeded in 96-well plates at 4000 cells per well and RADH85 EV/WT/Q* at 3000 cells per well and incubated in a 5% $CO_2$ humidified atmosphere at 37 °C for 24 h. U87 or RADH85 cells were then treated with CD95L (doses and times indicated in the figure legends) in the presence of 250 nM of a fluorescent dead cell marker (Incucyte® Cytotox Red Dye, #4632, Essen BioScience, Germany). RADH87 cells were pretreated with 200 nM (2X) birinapant during 1 h before adding 1 µg/mL of IgCD95L and Cytotox for 24 h. For cell death assays performed in the presence of MKC-8866, U87 WT were seeded in 96-well plates at 5,000 cells per well incubated in a 5% $CO_2$ humidified atmosphere at 37 °C for 24 h. Cells were pre-treated with DMSO or MKC-8866 (30 µM) and further treated with CD95L in the presence of 250 nM of Cytotox. Plates were placed in the Incucyte® (SX5 Live-Cell Analysis System, Essen BioScience, Germany). Both phase and fluorescent images were recorded (400 ms exposure, 10xlens). Four images were taken every 2 h. Each condition was recorded in triplicate. Images were analysed using the Adherent Cell-by-cell on Red phase algorithm and Cell-by-cell Classification of High Red Intensity (% of total cells) algorithm from Incucyte® 2022B Rev2 software.

## Flow cytometry

To evaluate CD95 expression at the cell surface, cells (U87, RADH85 or RADH87) were seeded in 6 well plates (250000 cells/

well). Twenty-four hours later, cells were harvested and stained using Zombie-violet™ (Biolegend, 423113) diluted in PBS according to the manufacturer protocol. Then, samples were saturated with PBS containing 1% BSA and 1% FCS and human FcR-blocking reagent (Miltenyi Biotec, 130-059-901) for 15 min at 4 °C prior to labelling using an anti-human CD95-APC antibody (clone DX2, Miltenyi Biotec, 130-117-701, AB_2751411) or corresponding IgG1-APC isotype (clone IS5-21F5, Miltenyi Biotec, 130-113-196, AB_2733440) for 30 min at 4 °C. Following two washes with the saturation buffer, cells were resuspended in PBS and analysed using a Novocyte cytometer (Acea Biosciences). CD95 cell surface expression was analysed after gating on viable singlets. Results are represented as a ratio of Median Intensity Fluorescence values for anti-CD95-stained over isotype-stained samples.

## Transfection

For knockdown experiments targeting CD95, U87 cells were seeded in six-well plates at 250,000 cells per well and transfected 24 h later using LipoRNAimax (Invitrogen, 13778075) according to the manufacturer protocol with 80 nM siRNA (On-target plus Non-targeting or Human FAS pool, Horizon Discovery, D-001810-10-05 or L-003776-00-0005, respectively). Twenty-four hours after transfection, cells were reseeded in 48-well plate and six-well plate for viability and western blot analysis respectively. Twenty-four hours after reseeding, cells were either lysed (for western blot) or treated as described for viability assay.

For over-expression of XBP1s, U87 or SUM159 cells seeded in six-well plates at 250,000 cells per well were transfected 24 h after seeding using 2.5 µg DNA per well and lipofectamine 3000 (Invitrogen, L3000008) according to the manufacturer protocol. pcDNA3.1 + -XBP1s-FLAG was obtained from Genescript (OHu25513D Accession No: NM_001079539.1, Vector: pcDNA3.1+/C-(K)DYK Species: Human). Twenty-four hours after transfection, cells were reseeded in 48-well plate. Twenty-four hours after reseeding, cells were treated as described for viability assay.

For short and long-term knockdown experiments targeting IRE1, PERK or XBP1, U87 WT cells were seeded in six-well plates at 150,000 cells per well and incubated in a 5% $CO_2$ humidified atmosphere at 37 °C for 24 h. The cells were then transfected using

LipoRNAimax (Invitrogen, 13778-075), following the supplier's recommendations. Specifically, 30 nM siRNA (Dharmacon TM On target plus SMART pool Human XBP1 (L-009552-00-0010); Dharmacon TM On target plus SMART pool Human EIF2AK3 (L-004883-00-0005); Dharmacon TM On-target plus TM Control pool Non-targeting pool (D-001810-10-05); Santa Cruz Biotechnology, IRE1α siRNA (h) (sc-40705) were used. 16 h post-transfection, cells were further treated with thapsigargin or tunicamycin for 8 h prior to cell lysis and analysis of protein extracts by western blot (Figs. 5A and EV4A). Alternatively, 72 h post-transfection cells were either lysed for protein analyses (Fig. 5E–G).

### Viability assay

Cells seeded in 48-well plates (20,000 cells per well) were treated as indicated in the figure legends 24 h post-seeding. 500 µg/mL MTT (Invitrogen, M6494) was added to the medium for the last two hours of treatment. Medium was then removed and the formazan crystals were solubilized using DMSO. Reading of the absorbance was performed at 560 nm using a Tecan Infinite F200 Pro reader. Results are expressed as % of viability, with 100% being untreated cells for each cell line or experimental condition.

### In vivo experiments

All in vivo experiments described in this study have been approved by the "Comité d'éthique de l'Université de Rennes 1" and the French ministry of education, research and innovation under the licence APAFIS #32241-2021061714545440. These experiments were performed at the ARCHE-BIOSIT UMS 34380 (Rennes). 8-week-old C57BL/6rJ male mice were obtained from Janvier. The animals were housed in groups in order to establish social relationships with free access to food and water and adapted bedding. The animals were kept under regulated light, temperature and humidity. Mice were fasted on day 1 at 12 pm for both experiments. Volumes injected intra-peritoneally did not exceed 250 µL for both experiments. For experiment 1, intra-peritoneal injections of MKC-8866 (or equivalent volume of DMSO) diluted in PBS to 1 mg/mL were performed at 6 pm on day 1 and 8 am on day 2 (10 mg/kg per injection) and mice were culled at 12 pm on day 2. For IHC analyses, a liver piece (around 200 mg) was fixed in paraformaldehyde 4% solution. The rest of the liver was rapidly cut in small pieces prior to flash-freezing using liquid nitrogen for western blot analysis. For experiment 2, intra-peritoneal injections of MKC-8866 (or equivalent volume of DMSO) diluted in PBS to 1 mg/mL were performed at 12 pm on day 1, 6 pm on day 1 and 8 am on day 2 (10 mg/kg per injection). At 12 pm on day 2, an intra-peritoneal injection of 0,15 mg/kg of anti-CD95 antibody Jo2 (BD Biosciences, 554254) or corresponding isotype control (BD Biosciences, 553961) was performed. Mice were culled 6 h later and liver samples collected as described for experiment 1.

### Statistical analyses and data representation

All statistical analyses were performed using Graphpad PRISM (v9) and are described in the figure legends. When performing tests with normality assumptions, the Shapiro-wilk test was used. For assumption of equality of variance, the Brown-Forsythe test or the *F*-test was used, as appropriate. Graphs were generated in Graphpad PRISM (v9) and figures were assembled using Adobe Illustrator.

### RNAseq and TCGA analyses

For each sample, the XBP1s and RIDD activity scores were calculated using the 38 genes from the IRE1 signature (Lhomond et al, 2018). The expression values of the 38 transcripts were extracted from the normalized count matrix of the 62 TNBC transcriptomes from a local cohort (GSE182021) and the 45 TCGA-GBM transcriptomes (See Appendix Table S2). Samples were then classified as high or low for XBP1s and RIDD activities according to the median. Then CD95, TNFR1, TRAIL-R1, TRAIL-R2, FADD or caspase-8 mRNA expression z-score was calculated for each sample and plotted according to the XBP1s and RIDD activities in both TNBC and GB datasets.

### Identification of potential XBP1s-binding sites of FAS murine and human promoter regions

Potential XBP1s-binding sites were searched for using TFBIND (Tsunoda and Takagi, 1999) within the 1500 bp region upstream of the ATG of murine and human *FAS* genes. CLUSTAL omega (Madeira et al, 2022) was also used to evaluate the extent of identity between mouse and human *FAS* promoter region sequences.

### Immunohistochemistry (IHC) and quantification

For IHC, samples were fixed in PFA 4% for at least 24 h, embedded in paraffin at least 12 h and sliced (4 µm) using a Leica microtome on Superfrost Plus slides (VWR, 631-0108) prior to drying at 60 °C for 1 h. The immunochemistry experiments were performed using the Discovery XT machine (Roche) and the Chromo-Map DAB kit (Roche). The following primary antibodies: cleaved caspase-3, Cell Signalling, 9661, AB_2341188, diluted 1/300; CD95, R&D systems, AF435, AB_355358 diluted 1/50 in antibody diluent (NB-23-00171-1, NeoBiotech) were incubated for 1 h at 37 °C. To perform the analysis, glass slides were digitized with the scanner Nanozoomer 2.0-RS Hamamatsu. The quantifications of CD95 expression and cleaved caspase-3 staining were carried out using the NIS-Elements software (Nikon) and averaged from 5 fields covering 4 mm² in total per liver.

## Data availability

This study includes no data deposited in external repositories. No particular blinding was used for conducting or analysing the experiments presented in this study.

## Peer review information

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

## Acknowledgements

We thank all the members of the U1242 for fruitful scientific discussions. We thank Dr Marc Aubry for his help with bioinformatic analyses. We thank Drs Michel Samson and Jacques Le Seyec for their advice on the in vivo liver damage model. We thank the BIOSIT H2P2 platform for immunohistochemistry, in particular Gevorg Ghukasyan, and the BIOSIT Animal facility ARCHE (https://biosit.univ-rennes1.fr/). This work was funded by grants from Fondation ARC (PDF20171206671) and Fondation de France to EL, European Union (EU) H2020 MSCA ITN-675448 (TRAINERS) and INCa PLBIO 2018, 2019 grants to EC, PJA20181207700 (Fondation ARC) grant to EL and MLG, Défis scientifiques from University of Rennes to EL and TA, Ligue contre le cancer grants (from committees 22, 35, 36, 56, 85) to TA, Associations la Vannetaise to EL and MLG, and la Josselinaise des femmes funding to MLG and Centre Eugène Marquis (EL, EC, FC, MLG, TA and SM). With financial support from ITMO Cancer of Aviesan within the framework of the 2021-2030 Cancer Control Strategy, on funds administered by Inserm.

## Author contributions

**Diana Pelizzari-Raymundo**: Formal analysis; Validation; Investigation; Visualization; Writing—review and editing. **Victoria Maltret**: Formal analysis; Investigation; Visualization; Writing—review and editing. **Manon Nivet**: Formal analysis; Investigation; Visualization; Writing—review and editing. **Raphael Pineau**: Investigation; Methodology; Writing—review and editing. **Alexandra Papaioannou**: Formal analysis; Investigation; Visualization; Writing—review and editing. **Xingchen Zhou**: Investigation; Writing—review and editing. **Flavie Caradec**: Investigation; Writing—review and editing. **Sophie Martin**: Investigation; Writing—review and editing. **Matthieu Le Gallo**: Formal analysis; Funding acquisition; Investigation; Visualization; Writing—review and editing. **Tony Avril**: Conceptualization; Supervision; Funding acquisition; Writing—review and editing. **Eric Chevet**: Conceptualization; Formal analysis; Supervision; Funding acquisition; Validation; Investigation; Visualization; Project administration; Writing—review and editing. **Elodie Lafont**: Conceptualization; Formal analysis; Supervision; Funding acquisition; Validation; Investigation; Visualization; Writing—original draft; Project administration; Writing—review and editing.

## Disclosure and competing interests statement

EC is a founder of Thabor Therapeutics. The authors do not declare any conflict of interest.

# Expanded View Figures

**Figure EV1. IRE1 represses CD95 expression in GB cells upon ER stress.**

(**A**) CD95 protein level was evaluated using western blot on lysates from the indicated cells. One representative experiment out of three independent experiments is presented. (**B**) U87 or SUM159 cells were pre-incubated for 1 h with 1 µg/mL of actinomycin D and further treated with 10 µM MKC-8866 for 1 h followed by 2 h treatment with 10 µM MG-132 as indicated. CD95 mRNA expression level, normalized to GAPDH, was expressed as fold of value obtained for control (actinomycin-only treated samples). Mean ± SEM, $n = 3–4$. Unpaired *t*-test (for comparing MG-132 and MG-132 + MKC-treated group), ****$p = 0.0003$. (**C,D**) U87 or SUM159 cells were pre-incubated for 1 h with 1 µg/mL of actinomycin D and further treated with 10 µM MKC-8866 or 10 µM Z4 for 1 h followed by 2 h treatment with 1µg/mL tunicamycin as indicated. CD95 mRNA expression level, normalized to GAPDH, was expressed as fold of value obtained for control (actinomycin-only treated samples). Mean ± SEM, $n = 3–4$. Unpaired *t*-test (for comparing TM and TM + MKC groups for (**C**) or TM and TM + Z4 groups for (**D**)), (**C**) *$p = 0.0461$ for U87, *$p = 0.0437$ for SUM159, (**D**) *$p = 0.0241$ for U87, (ns, $p = 0.0538$ for SUM159).

▶

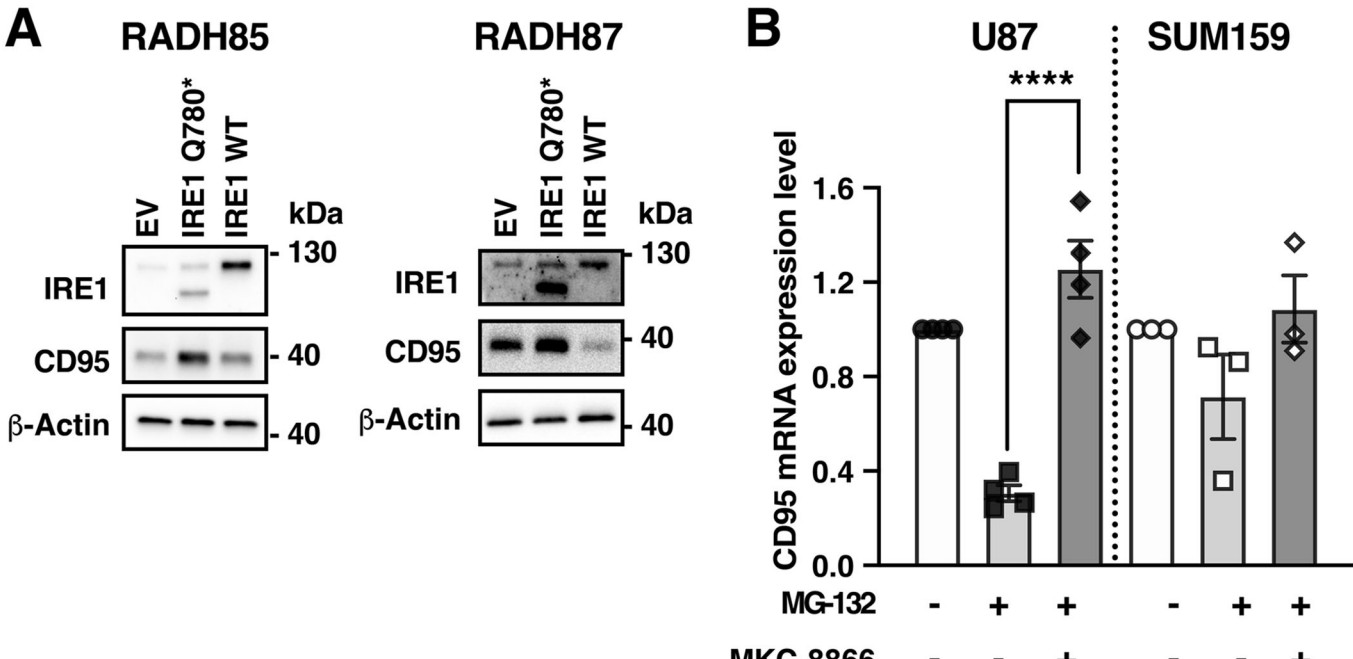

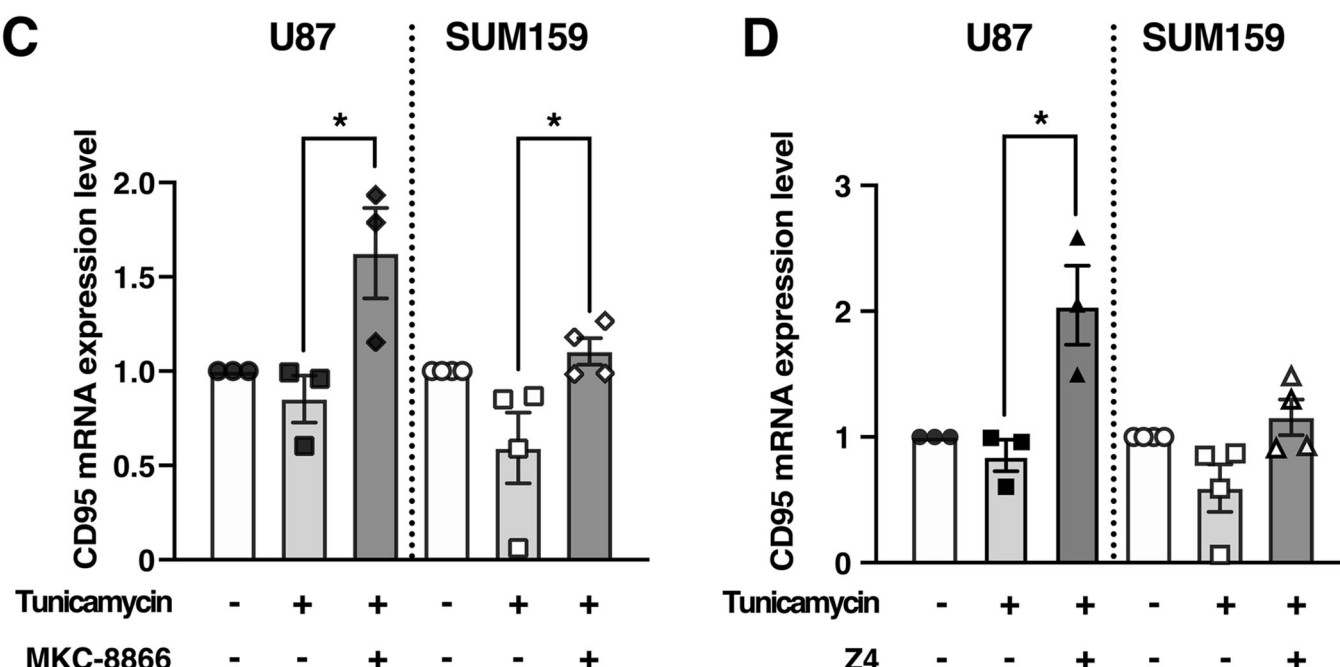

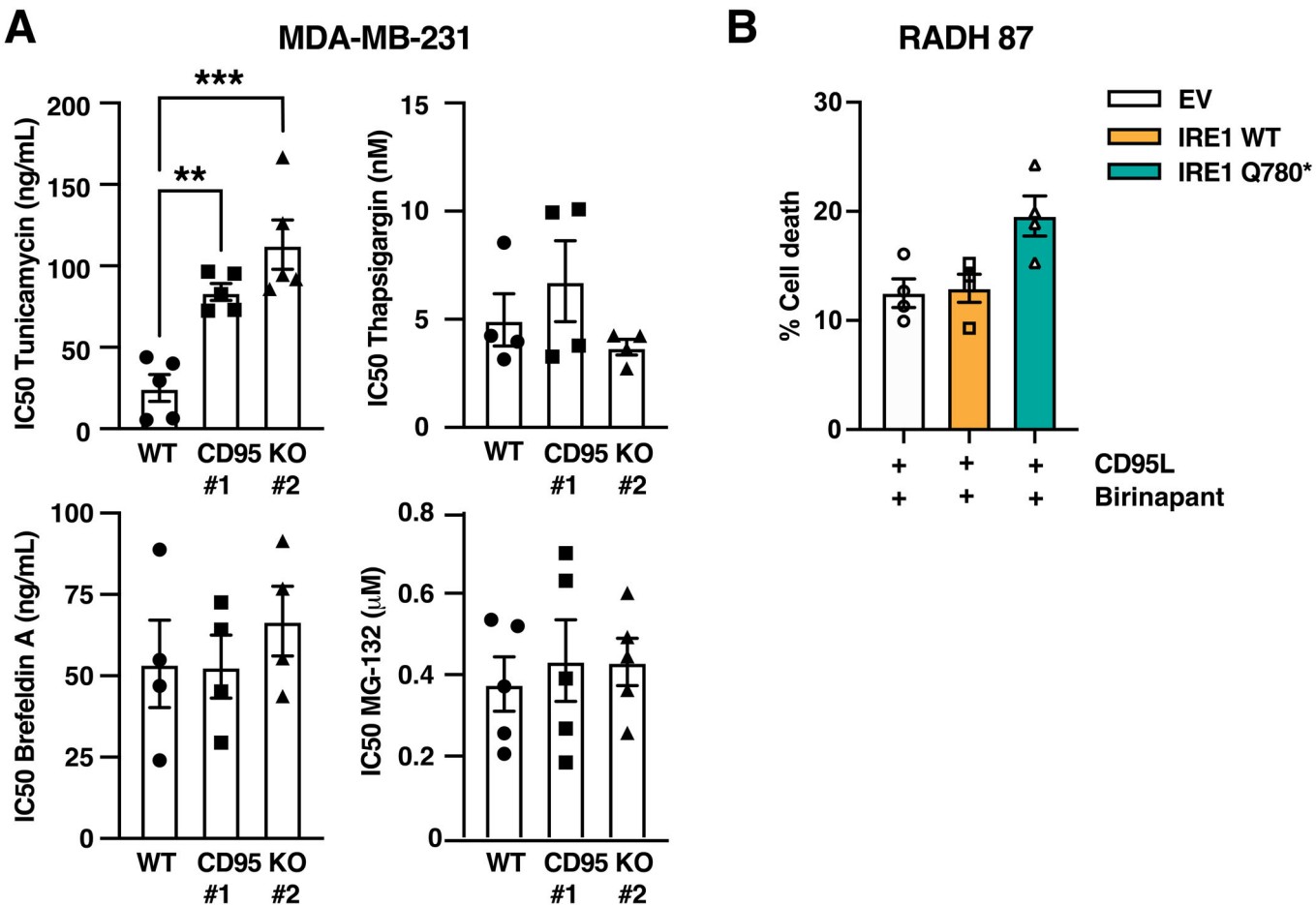

**Figure EV2. CD95 is not a universal determinant of ER-stress induced cell death whilst IRE1 RNase activity limits CD95L-induced cell death.**

(A) MDA-MB-231 WT or CD95 KO clones were treated for 48 h with the indicated ER stress inducers. Viability was determined using an MTT assay and relative IC50 calculated for each independent experiment (see also Appendix Fig. 2B). **$p = 0.044$, ***$p = 0.0002$, one-way ANOVA with Tukey multiple comparison correction. (B) RADH87 control (EV), stably expressing IRE1Q780* or IRE1WT were pre-treated with 200 nM (2X) of Birinapant for 1 h prior to addition of 1 µg/mL CD95L for 24 h. % of cell death was defined as the % of Cytotox red-positive cells as detected by the Incucyte. Mean ± SEM of three independent experiments.

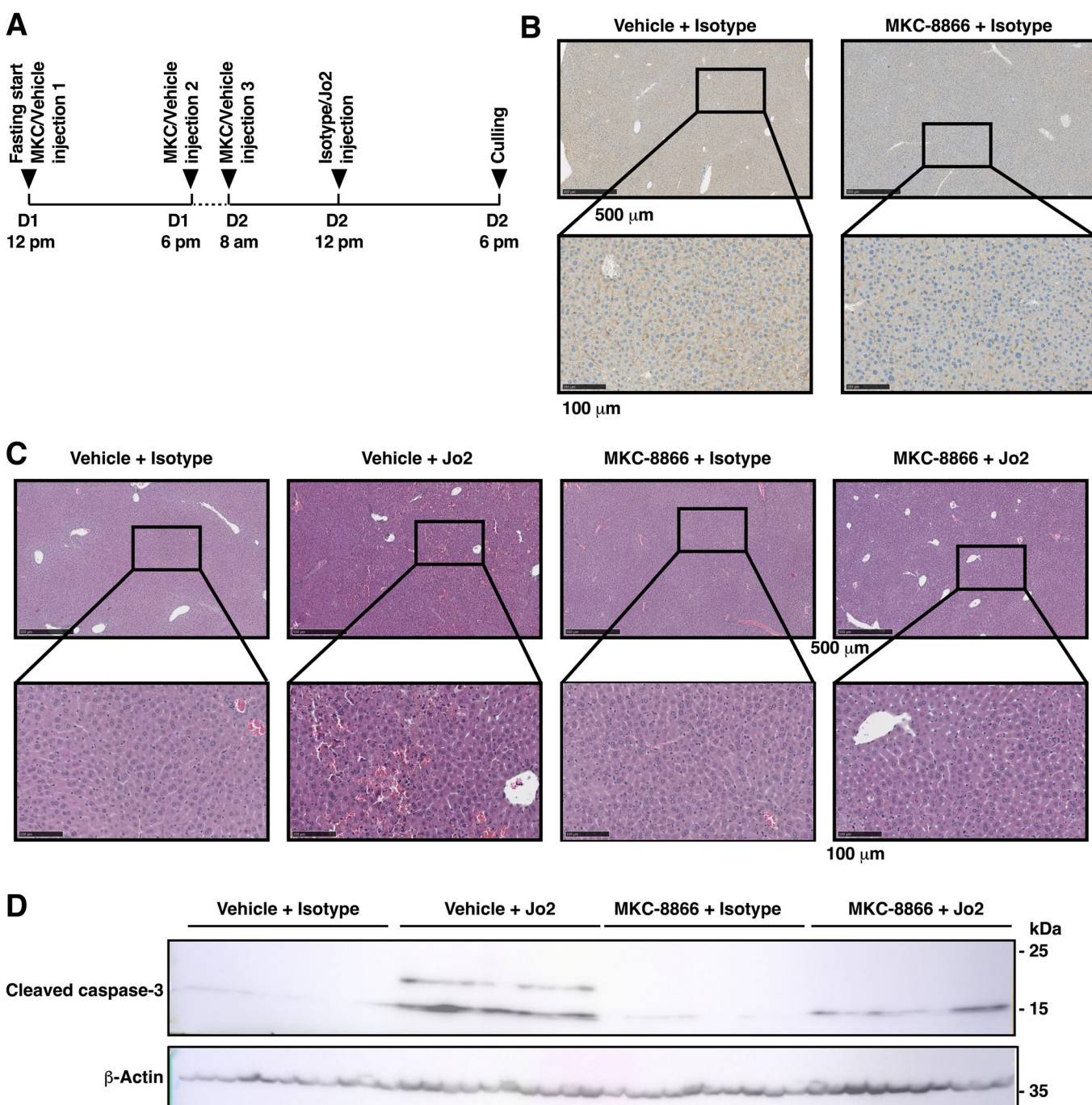

**Figure EV3. IRE1 RNase inhibition limits hepatic CD95 expression and CD95-mediated cell death in mice.**

(A) Timeline of the second in vivo experiment. 40 mice were divided in two groups of 20 and were repeatedly injected with either vehicle or MKC-8866 as indicated. On day 2 at 12 pm, each of this initial groups were further divided in two groups of 10 mice which were injected with either an anti-CD95 antibody or with an isotype control as indicated. (B) CD95 expression was evaluated by IHC in mice injected with vehicle or MKC-8866 and the isotype control antibody. One representative image is shown for each of these two groups. (C) HES staining was performed on liver tissue sections from mice of each of the four groups described in (A). One representative image is shown for each of these groups. (D) Western blot analysis of liver lysates from mice treated as indicated.

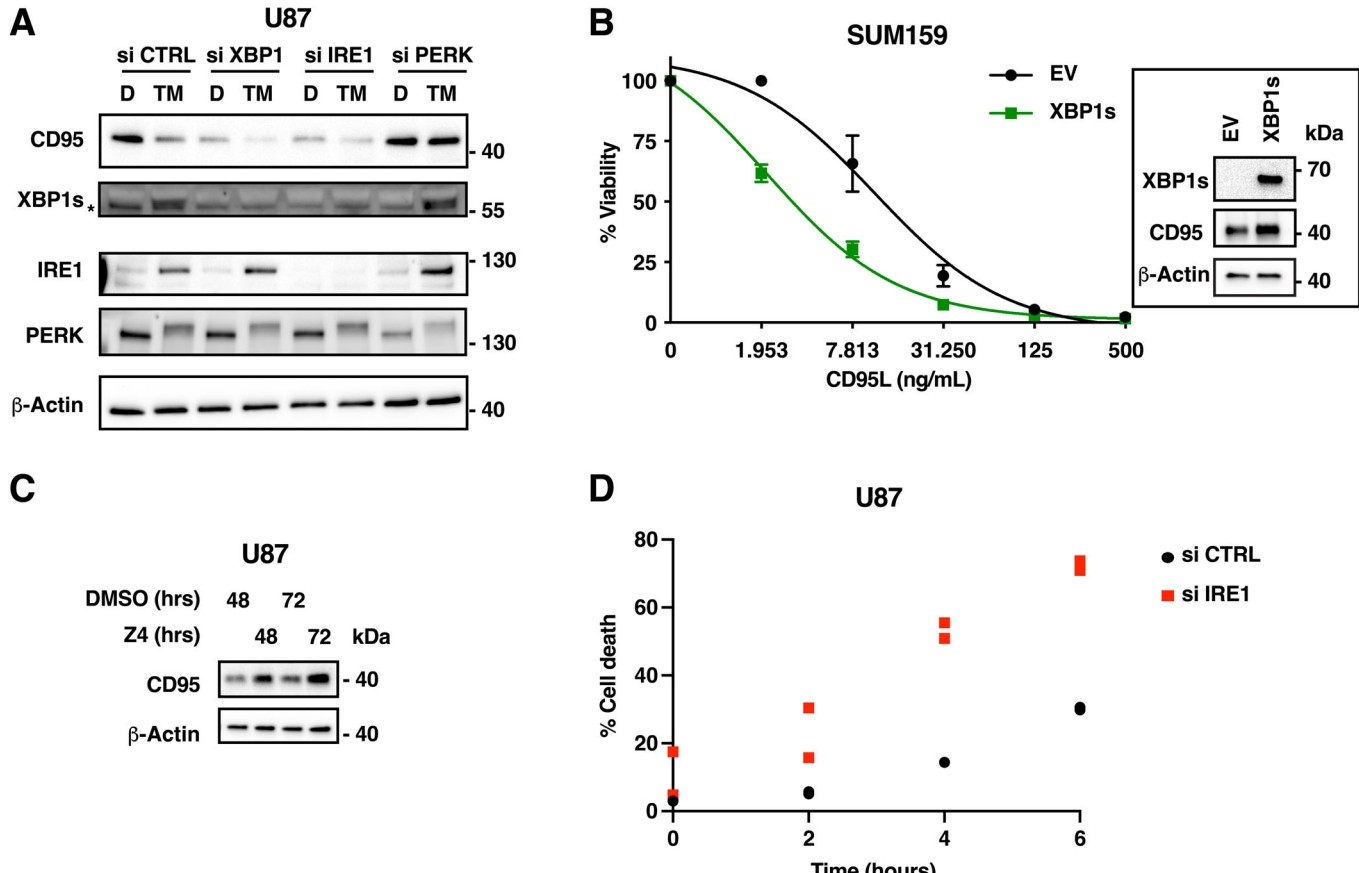

**Figure EV4. IRE1 RNase dually controls CD95 expression and CD95L-induced cell death.**

(A) U87 cells were transfected with siRNA control or targeting XBP1, IRE1 or PERK as indicated. 16 h later, cells were treated with 2.5 μg/mL thapsigargin for 8 h. Lysates were analysed using western blot. One experiment representative of at least three independent ones is shown. (B) SUM159 cells were transfected with a plasmid coding for FLAG-XBP1s (XBP1s) or an empty vector (EV). 48 h later, cells were treated with the indicated concentrations of CD95L for 48 h. Viability was assessed using MTT assay and normalized to untreated cell values. Mean ± SEM of three independent experiments. Inset: western blot analyses of cell lysates 48 h post-transfection. (C) U87 cells were treated with DMSO or Z4 (25 μM) for the indicated times. Lysates were analysed using western blot. One experiment representative of three independent ones is shown. (D) U87 were transfected with siRNA control or targeting IRE1 as indicated. 72 h later, cells were treated with 100 ng/mL CD95L. % of cell death was defined as the % of Cytotox red-positive cells as detected by the Incucyte. Two independent experiments are shown. Source data are available online for this figure.

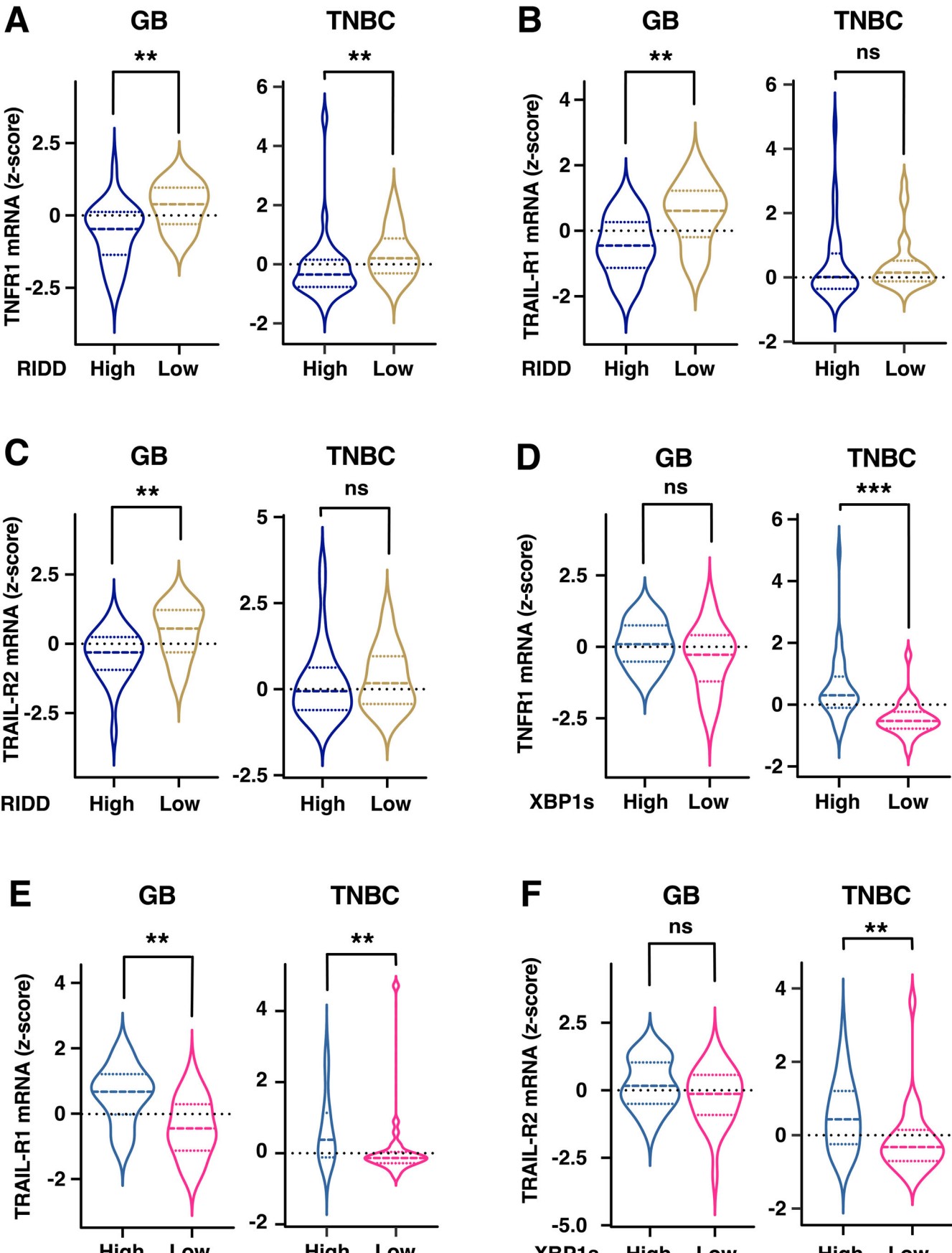

**Figure EV5. Low RIDD activity and high XBP1s activity in tumours correlate with the expression of DR.**

(A–F) TNFR1 (**A,D**), TRAIL-R1 (**B,E**), TRAIL-R2 (**C,F**) expression z-scores of 45 GB and 62 TNBC tumours were plotted according to the RIDD or XBP1 activity score. The distribution of z-score is represented as violin plots. For GB $n = 45$; for TNBC $n = 62$. Statistical difference of expression between groups was calculated using Mann–Whitney tests and the *p*-value is indicated ((**A**) **$p = 0.0016$ for GB and **$p = 0.0025$ for TNBC; (**B**) **$p = 0.0017$ for GB and $p = 0.13$ for TNBC; (**C**) **$p = 0.0044$ for GB and $p = 0.34$ for TNBC; (**D**) $p = 0.18$ for GB and ***$p = 2e - 06$ for TNBC; (**E**) **$p = 0.0011$ for GB and **$p = 0.0017$ for TNBC; (**F**) $p = 0.23$ for GB and **$p = 0.0016$ for TNBC).

