## [Peer Review File · EMBO Reports]

IRE1 RNase controls CD95-mediated cell death

Diana Pelizzari-Raymundo, Victoria Maltret, Manon Nivet, Raphael Pineau, Alexandra Papaioannou, Xingchen Zhou, Flavie Caradec, Sophie Martin, Matthieu Le Gallo, Tony Avril, Eric Chevet, and Elodie Lafont

Corresponding author(s): *Elodie Lafont* (elodie.lafont@inserm.fr)

Review Timeline:	Transfer from Review Commons:	31st Mar 23
	Editorial Decision:	13th Apr 23
	Revision Received:	12th Nov 23
	Editorial Decision:	12th Jan 24
	Revision Received:	18th Jan 24
	Accepted:	3rd Feb 24

Editor: *Bernd Pulverer*

**Transaction Report: This manuscript was transferred to
EMBO reports following peer review at Review Commons.**

**Review
COMMONS**

Review #1

1. Evidence, reproducibility and clarity:

Evidence, reproducibility and clarity (Required)

Summary:

Here the authors argue that IRE1 activation has opposite effects on Fas/CD95 expression/stability in a number of contexts, via either RIDD-dependent degradation of Fas mRNA or XBP-1-mediated induction of Fas expression, which led to either increased or decreased sensitivity to Fas-induced apoptosis in a number of settings.

Major issues:

The study is somewhat preliminary and inconclusive in that it is not clear why the RIDD function of XBP-1 appears to predominate *in vitro* in the cell lines examined, leading to modest increases in Fas expression levels (Figure 1) when IRE1 DN versus IRE1 WT constructs are overexpressed, which is at odds with the latter part of the paper which suggests that inhibition of RIDD led to reduced Fas expression levels. However, this could be due to supraphysiological levels of IRE1 being expressed under overexpression conditions, leading to confounding results. Similarly, when XBP-1s is overexpressed *in vitro* (Figure 5) the modest increases in CD95/Fas expression and sensitization to Fas-induced cell death may not be fully representative of what would be observed at physiological levels of XBP-1s activation. The *in vivo* results obtained using an IRE1 RNase inhibitor (MKC8866) contradict the earlier part of the study (as Fas levels decreased and there was protection from Fas-induced liver toxicity) and this could be due to a multitude of reasons. There is no doubt that impacting on IRE1 activity has interesting effects on CD95/Fas expression, which can be up- or -down-regulated, with consequences for cell death induced via engagement of the latter receptor, however, the manuscript does not offer a lot of clarity on which outcome is the predominant one in the context of engagement of the UPR. I have the following suggestions for improvement.

1. The authors should induce ER stress using Thaps, Brefeldin A and Tunicamycin, and explore the effects of doing this on Fas expression levels in the context of silencing endogenous IRE1, XBP-1 and PERK.
2. The authors should explore the effects of silencing of IRE1, XBP-1 and PERK on constitutive Fas expression and the outcome of Fas/CD95-induced apoptosis in cells not experiencing an overt activation of the UPR (i.e. in the absence of Thaps,

Brefeldin A or other UPR inducer).

3. The specificity of MKC8866 at the concentration used (30 uM) is unclear. What effect does MKCC have on sensitivity towards Fas-induced apoptosis, similar to the type of experimental set up presented in Figure 5A, 5B?

4. Similarly, what effects does MKC8866 (at 30 uM) have on key Fas pathway determinants, such as Fas, FLIPL, FLIPs, Caspase-8, FADD, RIPK1, A20, CYLD, cIAP-1, cIAP-2 and Bid? There are many points at which MKC8866 could influence the outcome of Fas receptor engagement beyond the receptor itself

****Minor issues:****

1. For the Fas mRNA cleavage experiments presented in Figure 2, there are no irrelevant control mRNAs to allow the reader to judge whether the effects presented are specific to Fas mRNA or are commonly observed for many mRNAs at these amounts of IRE1 (1 ug, 0.5 ug, which appear high).

2. Significance:

Significance (Required)

***General assessment:** this is an interesting study, as there is little knowledge currently concerning how the UPR influences Fas expression or Fas-dependent outcomes. However, the impact of this work is limited by the overexpression approaches used, which could produce artifactual results, as well as the contradictory message of the study.

***Advance:** the advance reported here is relatively modest and limited in scope due to the inconclusive nature of the data presented.

***Audience:** this study will be of interests to specialists in the UPR and cell death communities.

3. How much time do you estimate the authors will need to complete the suggested revisions:

Estimated time to Complete Revisions (Required)

(Decision Recommendation)

Between 3 and 6 months

Yes

Review #2

1. Evidence, reproducibility and clarity:

Evidence, reproducibility and clarity (Required)

The authors address here for the first time the connection between CD95, which is known as Fas, and ER stress. The role of another DR, TRAIL-R2 has been already reported, but this is the first study uncovering the link between Cd95 system and ER stress. The study is performed on the high level and supported by all necessary controls. They find the connection between IRE1 and CD95 and show that it might play a role in Cd95 signaling and attenuate CD95-mediated cell death.

Further, the correlation between CD95 expression and IRE is found in tumors. Importantly the authors find out the connection between XBP1 and CD95 expression, which was not reported to date. Hence, it is a very important and highly essential piece of research.

However, I would like to clarify the several issues:

1: Figure 1. Tunicamycin obviously leads to deglycosylation of CD95, which is indicated by the appearance of 35 Kda band. This should be highlighted and commented.

2. Figure 2c, d. The piece of mRNA structure, which is synthesized, might have the different secondary structure and might be not cleaved by IRE, accordingly. More

detailed comments have to be provided in this regard.

3. Figure 3. Caspase-8-3 western blots show beautiful effects but did authors see some effects further downstream, eg on PARP1 cleavage? Was cell death (not viability) measured as well? Can you comment on this?

4. Did the authors looked at the DISC assembly? Did they detect some differences there?

2. Significance:

Significance (Required)

This is an excellent study. The authors address here for the first time the connection between CD95, which is known as Fas, and ER stress. The role of another DR, TRAIL-R2 has been already reported, but this is the first study uncovering the link between Cd95 system and ER stress. The study is performed on the high level and supported by all necessary controls. This is an important advance for the death receptor field.

3. How much time do you estimate the authors will need to complete the suggested revisions:

Estimated time to Complete Revisions (Required)

(Decision Recommendation)

Between 1 and 3 months

Yes

Revision Plan

Manuscript number: RC-2022-01800

Corresponding author(s): Elodie Lafont

[The “revision plan” should delineate the revisions that authors intend to carry out in response to the points raised by the referees. It also provides the authors with the opportunity to explain their view of the paper and of the referee reports.]

The document is important for the editors of affiliate journals when they make a first decision on the transferred manuscript. It will also be useful to readers of the reprint and help them to obtain a balanced view of the paper.

*If you wish to submit a full revision, please use our "Full Revision" template. **It is important to use the appropriate template to clearly inform the editors of your intentions.**]*

1. General Statements [optional]

This section is optional. Insert here any general statements you wish to make about the goal of the study or about the reviews.

Several studies have previously demonstrated functional links between the death receptors (DR) TRAIL-R1/2 and the Unfolded protein response (UPR). In this manuscript, we describe the previously unrecognized IRE1-dependent dual regulation of the expression of another DR, CD95, and CD95L-induced cell death. Our work therefore adds to the current knowledge on the functional links existing between UPR and DR signaling and provides novel mechanistic insights on a dual regulation involving both transcriptional and post-transcriptional control of the expression of CD95 mRNA expression by IRE1. To demonstrate this, we have used both genetic (overexpression of XBP1s or dominant-negative forms of IRE1) and pharmacologic (IRE1 RNase inhibitor) approaches and cellular models of glioblastoma (GB) and triple-negative breast cancer (TNBC). We show that IRE1 RNase activity promotes CD95 expression and CD95-mediated cell death via the transcription factor XBP1s whilst IRE1 RNase limits CD95 expression and cell death via its ability to cleave RNAs (through RIDD, for Regulated IRE1-dependent decay of RNAs, activity). Furthermore, we report that IRE1-mediated control of CD95 expression is active *in vivo*, using a model of CD95-mediated fulminant hepatitis in mice. Lastly, we correlate these results to the pathology by showing that CD95 expression is decreased in RIDD high or XBP1s low human GB and TNBC tumors.

We thank the reviewers for their fair assessment of our manuscript and for their insightful comments. Below, we describe the experiments we plan to carry out to address the reviewers' comments.

2. Description of the planned revisions

Insert here a point-by-point reply that explains what revisions, additional experimentations and analyses are planned to address the points raised by the referees.

Reviewer #1 (Evidence, reproducibility and clarity (Required)):

Summary:

Here the authors argue that IRE1 activation has opposite effects on Fas/CD95 expression/stability in a number of contexts, via either RIDD-dependent degradation of Fas mRNA or XBP-1-mediated induction of Fas expression, which led to either increased or decreased sensitivity to Fas-induced apoptosis in a number of settings.

Major issues:

The study is somewhat preliminary and inconclusive in that it is not clear why the RIDD function of XBP-1 appears to predominate *in vitro* in the cell lines examined, leading to modest increases in Fas expression levels (Figure 1) when IRE1 DN versus IRE1 WT constructs are overexpressed, which is at odds with the latter part of the paper which suggests that inhibition of RIDD led to reduced Fas expression levels. However, this could be due to supraphysiological levels of IRE1 being expressed under overexpression conditions, leading to confounding results. Similarly, when XBP-1s is overexpressed *in vitro* (Figure 5) the modest increases in CD95/Fas expression and sensitization to Fas-induced cell death may not be fully representative of what would be observed at physiological levels of XBP-1s activation. The *in vivo* results obtained using an IRE1 RNase inhibitor (MKC8866) contradict the earlier part of the study (as Fas levels decreased and there was protection from Fas-induced liver toxicity) and this could be due to a multitude of reasons. There is no doubt that impacting on IRE1 activity has interesting effects on CD95/Fas expression, which can be up- or -down-regulated, with consequences for cell death induced via engagement of the latter receptor, however, the manuscript does not offer a lot of clarity on which outcome is the predominant one in the context of engagement of the UPR. I have the following suggestions for improvement.

We thank the reviewer for this overall positive assessment.

1. The authors should induce ER stress using Thaps, Brefeldin A and Tunicamycin, and explore the effects of doing this on Fas expression levels in the context of silencing endogenous IRE1, XBP-1 and PERK.

We do agree with this reviewer that the proposed experiments might further highlight which of the IRE1-dependent control of CD95 expression dominates upon ER stress induction. Therefore, we will perform the requested experiment in the various cell lines already used in the manuscript.

We propose to evaluate the expression of CD95 (at the mRNA and total protein levels) under ER stress induction (by different ER stressors) upon knock-down of IRE1 or XBP1. Other DRs (TRAIL-R1 and 2) have been shown to be induced by PERK activation and it is also demonstrated that PERK and IRE1 signaling pathways coregulate each other. As such, we also propose to assess whether PERK could also control CD95 expression in this setting.

2. The authors should explore the effects of silencing of IRE1, XBP-1 and PERK on constitutive Fas expression and the outcome of Fas/CD95-induced apoptosis in cells not experiencing an overt activation of the UPR (i.e. in the absence of Thaps, Brefeldin A or other UPR inducer).

We thank the reviewer for their suggestion and will perform the requested experiments as proposed.

3. The specificity of MKC8866 at the concentration used (30 μ M) is unclear. What effect does

MKCC have on sensitivity towards Fas-induced apoptosis, similar to the type of experimental set up presented in Figure 5A, 5B?

Regarding the specificity of MKC8866, this drug has been optimized and refined from a family of IRE1-specific endoribonuclease inhibitors initially obtained from a chemical library screen [1-3]. This salicylaldehyde analog has already shown to be effective in multiple cancer models including breast [4, 5] and prostate [2] cancers. We have recently demonstrated its efficacy in a GB mouse model [6]. It is therefore a widely used IRE1 inhibitor, including in the dose range 10-30 μ M used in this study (e.g [4, 5]). We therefore do not think it is in the scope of this manuscript to re-assess its specificity. However, we will aim at testing an additional IRE1 inhibitor to assess whether similar effects can be observed on CD95 expression in cells. To do so, we propose to use a novel IRE1 kinase inhibitor developed in the laboratory (DOI: 10.26434/chemrxiv-2022-2ld35 – Accepted iScience) and shown to efficiently blunt IRE1 activity in GB. As also suggested by the reviewer, we will assess whether the use of MKC-8866 can affect CD95L-induced cell death in cell lines.

4. Similarly, what effects does MKC8866 (at 30 μ M) have on key Fas pathway determinants, such as Fas, FLIPL, FLIPs, Caspase-8, FADD, RIPK1, A20, CYLD, cIAP-1, cIAP-2 and Bid? There are many points at which MKC8866 could influence the outcome of Fas receptor engagement beyond the receptor itself.

In the present manuscript, we have shown that MKC-8866 reduces CD95 expression in mouse liver (IHC depicted in Figure 4B and S3B) *in vivo* and that, when used at 30 μ M *in vitro*, it prevents the loss of CD95 expression induced by tunicamycin or thapsigargin in U87 cells (Fig 1C-F). We do agree with the reviewer that IRE1 may impact CD95-induced cell death beyond modulating CD95 expression, as also already discussed in the present manuscript. Therefore, and as suggested, we will assess whether MKC-8866, used at 30 μ M, also impacts on the basal cellular expression of the various components of CD95 signaling mentioned by this reviewer.

Minor issues:

1. For the Fas mRNA cleavage experiments presented in Figure 2, there are no irrelevant control mRNAs to allow the reader to judge whether the effects presented are specific to Fas mRNA or are commonly observed for many mRNAs at these amounts of IRE1 (1 μ g, 0.5 μ g, which appear high).

The expression of Fas mRNA was already normalized to GAPDH (which does not seem to vary upon incubation with IRE1). We nevertheless will test the expression of additional “irrelevant” RNAs as suggested by the reviewer.

Reviewer #1 (Significance (Required)):

General assessment: this is an interesting study, as there is little knowledge currently concerning how the UPR influences Fas expression or Fas-dependent outcomes. However, the impact of this work is limited by the overexpression approaches used, which could produce artifactual results, as well as the contradictory message of the study.

Although we think that the message of the manuscript is indeed complex, the work presented herein does not rely exclusively on overexpression approaches as our genetic-based results are also comforted by the use of pharmacologic inhibitors of IRE1.

Advance: the advance reported here is relatively modest and limited in scope due to the inconclusive nature of the data presented.

Audience: this study will be of interests to specialists in the UPR and cell death communities.

We thank the reviewer for acknowledging the overall novelty of our work. We do hope that the experiments proposed will address her/his remaining concerns.

Reviewer #2 (Evidence, reproducibility and clarity (Required)):

The authors address here for the first time the connection between CD95, which is known as Fas, and ER stress. The role of another DR, TRAIL-R2 has been already reported, but this is the first study uncovering the link between Cd95 system and ER stress. The study is performed on the high level and supported by all necessary controls. They find the connection between IRE1 and CD95 and show that it might play a role in Cd95 signaling and attenuate CD95-mediated cell death.

Further, the correlation between CD95 expression and IRE is found in tumors. Importantly the authors find out the connection between XBP1 and CD95 expression, which was not reported to date. Hence, it is a very important and highly essential piece of research.

We thank the reviewer for these very positive comments and the acknowledging of the novelty and importance of our study.

However, I would like to clarify the several issues:

1: Figure 1. Tunicamycin obviously leads to deglycosylation of CD95, which is indicated by the appearance of 35 Kda band. This should be highlighted and commented.

We agree, this will be commented on in the text.

2. Figure 2c, d. The piece of mRNA structure, which is synthesized, might have the different secondary structure and might be not cleaved by IRE, accordingly. More detailed comments have to be provided in this regard.

The model depicted in Figure 2B is a predicted computational secondary structure of CD95 mRNA. In the experiments performed in Figure 2A, C and D the mRNA was extracted from U87 cells prior to incubation with recombinant IRE1 and the resulting products analyzed using RT-qPCR with primers flanking different portions of the CD95 mRNA sequence. For Figures 2C and D, the primers used flank the two sites which were predicted to be cleaved by IRE1 based on previous work from our lab [7]. Even though we cannot exclude that additional sites can be targeted beyond these two, the fact that the amplification of CD95 sequence is reduced in samples pre-incubated with recombinant IRE1 strongly suggests that IRE1 is indeed able to cleave CD95 mRNA in these regions in vitro. We will modify the main text to further explain this point.

3. Figure 3. Caspase-8-3 western blots show beautiful effects but did authors see some effects further downstream, eg on PARP1 cleavage? Was cell death (not viability) measured as well? Can you comment on this?

This is absolutely right, we will test PARP-1 cleavage in this setting as suggested. Given the morphology of the cells we observed in the viability experiments, we would expect a similar trend using cell death assays. However, we do agree with the reviewer that this should be proven experimentally, so we will perform these experiments again using cell death assays as a read out.

4. Did the authors look at the DISC assembly? Did they detect some differences there? **No, we did not. We would expect some difference given the impact we have observed on CD95 expression, caspase-8 activation and cell death of expressing dominant negative forms of IRE1, but this of course needs to be actually tested. We are in the process of optimizing CD95 DISC experiments in our lab and we therefore hope to be able to address this reviewer's comment in a revised version of the manuscript.**

Reviewer #2 (Significance (Required)):

This is an excellent study. The authors address here for the first time the connection between CD95, which is known as Fas, and ER stress. The role of another DR, TRAIL-R2 has been already reported, but this is the first study uncovering the link between Cd95 system and ER stress. The study is performed on the high level and supported by all necessary controls. This is an important advance for the death receptor field.

Thank you again for these very positive comments and your insightful appreciation of our work.

3. Description of the revisions that have already been incorporated in the transferred manuscript

Please insert a point-by-point reply describing the revisions that were already carried out and included in the transferred manuscript. If no revisions have been carried out yet, please leave this section empty.

N/A

4. Description of analyses that authors prefer not to carry out

Please include a point-by-point response explaining why some of the requested data or additional analyses might not be necessary or cannot be provided within the scope of a revision. This can be due to time or resource limitations or in case of disagreement about the necessity of such additional data given the scope of the study. Please leave empty if not applicable.

As explained in our response to point 3 of reviewer 1, we think that re-demonstrating the specificity of MKC-8866 is beyond the scope of the present study. We will use an additional IRE1 inhibitor in vitro as an alternative to MKC-8866 (in addition to performing the siRNA-based experiments suggested by the reviewer).

References

1. Volkman, K., Lucas, J. L., Vuga, D., Wang, X., Brumm, D., Stiles, C., Kriebel, D., Der-Sarkissian, A., Krishnan, K., Schweitzer, C., Liu, Z., Malyankar, U. M., Chiovitti, D., Canny, M., Durocher, D., Sicheri, F. & Patterson, J. B. (2011) Potent and selective inhibitors of the inositol-requiring enzyme 1 endoribonuclease, *J Biol Chem.* **286**, 12743-55.
2. Sheng, X., Nenseth, H. Z., Qu, S., Kuzu, O. F., Frahnaw, T., Simon, L., Greene, S., Zeng, Q., Fazli, L., Rennie, P. S., Mills, I. G., Danielsen, H., Theis, F., Patterson, J. B., Jin, Y. & Saatcioglu, F. (2019) IRE1 α -XBP1s pathway promotes prostate cancer by activating c-MYC signaling, *Nat Commun.* **10**, 323.

3. Langlais, T., Pelizzari-Raymundo, D., Mahdizadeh, S. J., Gouault, N., Carreaux, F., Chevet, E., Eriksson, L. A. & Guillory, X. (2021) Structural and molecular bases to IRE1 activity modulation, *Biochem J.* **478**, 2953-2975.
4. Logue, S. E., McGrath, E. P., Cleary, P., Greene, S., Mnich, K., Almanza, A., Chevet, E., Dwyer, R. M., Oommen, A., Legembre, P., Godey, F., Madden, E. C., Leuzzi, B., Obacz, J., Zeng, Q., Patterson, J. B., Jager, R., Gorman, A. M. & Samali, A. (2018) Inhibition of IRE1 RNase activity modulates the tumor cell secretome and enhances response to chemotherapy, *Nat Commun.* **9**, 3267.
5. Almanza, A., Mnich, K., Blomme, A., Robinson, C. M., Rodriguez-Blanco, G., Kierszniowska, S., McGrath, E. P., Le Gallo, M., Pilalis, E., Swinnen, J. V., Chatziioannou, A., Chevet, E., Gorman, A. M. & Samali, A. (2022) Regulated IRE1 α -dependent decay (RIDD)-mediated reprogramming of lipid metabolism in cancer, *Nat Commun.* **13**, 2493.
6. Le Reste, P. J., Pineau, R., Voutetakis, K., Samal, J., Jégou, G., Lhomond, S., Gorman, A. M., Samali, A., Patterson, J. B., Zeng, Q., Pandit, A., Aubry, M., Soriano, N., Etcheverry, A., Chatziioannou, A., Mosser, J., Avril, T. & Chevet, E. (2020) Local intracerebral Inhibition of IRE1 by MKC8866 sensitizes glioblastoma to irradiation/chemotherapy in vivo, 841296.
7. Voutetakis, K. D., D.; Vlachavas, E-I., Leonidas, DD.; Chevet, E.; Chatziioannou, A. (In preparation) RNA sequence motif and structure in IRE1-mediated cleavage.

Dear Dr. Lafont

Thank you for the submission of your manuscript to EMBO reports after peer review at Review Commons. We have now reviewed the manuscript as well as the two referee reports and your detailed response and revision plan.

We acknowledge the conceptual interest of the work. However, we also note that while referee 2 is quite positive about the interest of the dataset for a broad readership, referee 1, a very experienced subject expert used frequently by the journal as a referee, is much more circumspect about the impact of the dataset - at least in its current form.

Nonetheless, your revision plan shows that will certainly work to address all specific referee points experimentally. It seems reasonable to us to propose to use another inhibitor to address point #3 of referee 1.

We therefore propose that we will return a manuscript revised according to your revision plan in due course to the current referees. Given the diverse assessment by both referees, we may send the revision and referee reports to a third independent expert in an arbitrating function (i.e. not as a de novo third referee) if we feel this would help provide a more balanced set of views.

We realize that it is difficult to revise to a specific deadline. In the interest of protecting the conceptual advance provided by the work, we recommend a revision within 3 months (14th Jul 2023). Please discuss the revision progress ahead of this time with the editor if you require more time to complete the revisions.

- 1) A data availability section providing access to data deposited in public databases is missing. If you have not deposited any data, please add a sentence to the data availability section that explains that.
- 2) Your manuscript contains statistics and error bars based on $n=2$. Please use scatter blots in these cases. No statistics should be calculated if $n=2$.

5) a complete author checklist, which you can download from our author guidelines . Please insert information in the checklist that is also reflected in the manuscript. The completed author checklist will also be part of the RPF.

6) Please note that all corresponding authors are required to supply an ORCID ID for their name upon submission of a revised manuscript (). Please find instructions on how to link your ORCID ID to your account in our manuscript tracking system in our

Author guidelines

- the name of the statistical test used to generate error bars and P values,
- the number (n) of independent experiments (please specify technical or biological replicates) underlying each data point,
- the nature of the bars and error bars (s.d., s.e.m.),
- If the data are obtained from n Program fragment delivered error ``Can't locate object method "less" via package "than" (perhaps you forgot to load "than"?) at //ejpvfs23/sites23b/embor_www/letters/embor_decision_rc_revise_and_rereview.txt line 56.' 2, use scatter blots showing the individual data points.

I look forward to seeing a revised form of your manuscript when it is ready.

Yours sincerely,

Bernd Pulverer

~~~~~  
Bernd Pulverer, Ph.D.  
Chief Editor, EMBO Reports  
EMBO  
Meyerhofstrasse 1, D-69117 Heidelberg  
Tel: +4962218891501  
bernd.pulverer@embo.org  
~~~~~

EMBO Reports**Manuscript number:** RC-2022-01800*In black, the reviewer comments.**In blue, the authors' responses.***Reviewer #1 (Evidence, reproducibility and clarity (Required)):**

Summary: Here the authors argue that IRE1 activation has opposite effects on Fas/CD95 expression/stability in a number of contexts, via either RIDD-dependent degradation of Fas mRNA or XBP-1-mediated induction of Fas expression, which led to either increased or decreased sensitivity to Fas-induced apoptosis in a number of settings.

Major issues:

The study is somewhat preliminary and inconclusive in that it is not clear why the RIDD function of XBP-1 appears to predominate in vitro in the cell lines examined, leading to modest increases in Fas expression levels (Figure 1) when IRE1 DN versus IRE1 WT constructs are overexpressed, which is at odds with the latter part of the paper which suggests that inhibition of RIDD led to reduced Fas expression levels. However, this could be due to supraphysiological levels of IRE1 being expressed under overexpression conditions, leading to confounding results. Similarly, when XBP-1s is overexpressed in vitro (Figure 5) the modest increases in CD95/Fas expression and sensitization to Fas-induced cell death may not be fully representative of what would be observed at physiological levels of XBP-1s activation. The in vivo results obtained using an IRE1 RNase inhibitor (MKC8866) contradict the earlier part of the study (as Fas levels decreased and there was protection from Fas-induced liver toxicity) and this could be due to a multitude of reasons. There is no doubt that impacting on IRE1 activity has interesting effects on CD95/Fas expression, which can be up- or -down-regulated, with consequences for cell death induced via engagement of the latter receptor, however, the manuscript does not offer a lot of clarity on which outcome is the predominant one in the context of engagement of the UPR. I have the following suggestions for improvement.

We thank reviewer #1 for their assessment. We have now addressed their comments in full and we believe this has vastly improved the quality of this study and the strength of its main messages, as further detailed below.

*As advised, we have now used several **genetic** (including transient knockdowns (siRNA) of XBP1 or IRE1, overexpression of XBP1s or dominant-negative forms of IRE1) and **pharmacologic** (one IRE1 RNase and one IRE1 kinase inhibitor) approaches in cellular models of glioblastoma (GB) and triple-negative breast cancer (TNBC). The data obtained now clearly show that IRE1 RNase activity promotes CD95 expression via the transcription factor XBP1s and CD95-mediated cell death whilst IRE1 RNase limits CD95 expression and cell death via its ability to cleave RNAs (through RIDD, for Regulated IRE1-dependent decay of RNAs, activity).*

*IRE1-mediated control of CD95 expression is also active *in vivo*, as demonstrated using a model of CD95-mediated fulminant hepatitis in mice and the pharmacologic inhibition of IRE1 by the IRE1 RNase inhibitor MKC-8866.*

Lastly, we correlate these results with those obtained in patients' brain and breast tumors (pathological context) by showing that CD95 mRNA expression is decreased in RIDD high or XBP1s low human GB and Triple Negative Breast Cancer tumors (compared to their low RIDD and high XBP1s counterparts). This study therefore demonstrates a previously unrecognized IRE1-dependent dual regulation of the expression of CD95 and CD95L-induced

cell death. The precise understanding of the events which govern the dominance of XBP1s or RIDD (perhaps RIDDLE too) on CD95 expression (but also likely on other mRNAs which behave in a similar manner) remains to be elucidated and for this we believe we will benefit from the community efforts trying to characterize the molecular mechanisms underlying this dichotomy. We think that, at present stage, this is beyond the scope of the current manuscript.

Overall, our work not only adds to the current knowledge on the functional links existing between UPR and DR signaling but also provides novel mechanistic insights on a dual regulation involving both transcriptional and post-transcriptional control of the expression of CD95 mRNA expression by IRE1.

1. The authors should induce ER stress using Thaps, Brefeldin A and Tunicamycin, and explore the effects of doing this on Fas expression levels in the context of silencing endogenous IRE1, XBP-1 and PERK.

We have now performed the requested experiments, using thapsigargin and tunicamycin (we did not use BFA which alters directly the traffic of ATF6 to the Golgi complex and its cleavage-mediated activation. This might lead to altered XBP1u expression and therefore to altered kinetics of XBP1s expression and could yield confounding effects) following knockdown of IRE1 or XBP1 in U87. Since other DRs (TRAIL-R1 and 2) have been shown to be induced by PERK activation (Lafont, 2020) and since PERK can also repress IRE1 activation (Chang *et al*, 2018) or promote its expression (Tsuru *et al*, 2016) we have also evaluated the expression of CD95 under ER stress induction upon knock-down of PERK as suggested by this reviewer.

The results newly obtained demonstrate that upon acute ER stress induction (8 hrs 2.5 µg/mL tunicamycin or 250 nM thapsigargin), CD95 expression is reduced. This decrease is IRE1-, but not XBP1- dependent since it is repressed by knockdown of IRE1 but not by that of XBP1 (**new Figures 5A and EV4A**). Interestingly, knockdown of PERK also limited ER stress-induced decrease of CD95 protein expression without majorly affecting IRE1 levels. We therefore hypothesize that PERK may either promote IRE1 RIDD activity -and thus promote acute ER stress-induced CD95 expression decrease- by a yet undefined mechanism. Alternatively (and probably more likely), silencing of PERK may counteract ER stress-induced decrease of CD95 by preventing ER-stress induced protein synthesis attenuation. This is now commented on in the main text (pages 6-7).

New Figures EV4A (left), and 5A (right). U87 were transfected with siRNA control or targeting XBP1, IRE1 or PERK as indicated. 16 hours later, cells were treated with DMSO (D), 2.5 mg/mL tunicamycin (TM) or 250 nM thapsigargin (TG) for 8 hours. Lysates were analysed by western blot. One experiment representative of at least three independent ones is shown. *lower band is not specific.

2. The authors should explore the effects of silencing of IRE1, XBP-1 and PERK on constitutive Fas expression and the outcome of Fas/CD95-induced apoptosis in cells not experiencing an overt activation of the UPR (i.e. in the absence of Thaps, Brefeldin A or other UPR inducer).

We thank the reviewer for this suggestion and we have now performed the requested experiment. As shown in **Figures 5A and EV4A** ('DMSO' conditions, 24 hrs siRNA transfection) and new **Figures 5E-G** (72 hrs siRNA transfection), knockdown of XBP1 consistently represses CD95 expression in U87 cells. Along with our initial observation that overexpression of XBP1s in SUM159 or U87 cells promotes CD95 expression (**Figure 5B and EV4B**) and that MKC-8866 reduces hepatic CD95 expression in vivo (**Figure 4B and EV3B**), these results strongly suggest that CD95 is indeed a *bona fide* XBP1s target. Whilst short-term knockdown of IRE1 also represses CD95 expression, its long-term depletion promotes CD95 expression (**new Figures 5A, EV4A, se above and 5E-G**). Therefore, we hypothesize that the constitutive expression of CD95 is predominantly induced by the XBP1s branch in the cellular models used. The dual impact of knocking-down IRE1 on CD95 expression, which reveals the dominance of the XBP1s branch at early time points and of the RIDD branch at late time points of transfection, likely represents the result of a cellular adaptation of the cells to the stressful condition of transfection (in a cancer context, it might also yield to a selection process). Similarly, our initial observation using dominant-negative versus WT IRE1-expressing cells (**Figure 1A, B**), which showed a RIDD dominance, also likely constituted a chronically stressed/adapted cellular model.

Taken together with the results obtained with the pharmacologic inhibition of IRE1 (see below), our results suggest that the XBP1s branch is dominant in controlling the constitutive/physiologic expression of CD95 whilst the RIDD is dominant upon acute/intense ER stress induction or prolonged/chronic stress. Interestingly, our results also show that in long-term transfection conditions, PERK contributes to CD95 expression too (**new Figure 5F**), indicating that this DR expression is in fact tightly regulated by multiple ER stress sensors.

New Figures 5 E-G and new Figure EV4D. **5E-G.** U87 were transfected with siRNA control or targeting XBP1, IRE1 or PERK as indicated. 72 hours later, cell lysates were analysed by western blot. One experiment representative of at least three independent ones is shown. **EV4D.** U87 were transfected with siRNA control or targeting IRE1 as indicated. 72 hours later, cells were treated with 100 ng/mL CD95L. % of cell death was defined as the % of Cytotox red-positive cells as detected by the Incucyte. Two independent experiments are shown (mean with range).

With regard to cell death, long-term depletion of IRE1 sensitized cells to CD95L-induced cell death, in line with enhanced CD95 expression (**new Figure EV4D**). As shown below (**reviewer-only figure 1**), knockdown of PERK or XBP1 also sensitized the cells, albeit to a much lesser extent. Whilst this latter result may seem at odds with the level of expression of CD95, one may consider that CD95 is not the sole determinant of sensitivity to CD95L and the long-term knockdown of XBP1 or PERK may have affected other components of the pathway. For

knockdown of XBP1, we also cannot exclude that by targeting XBP1u as well, the siRNA might have additional effect in the long-term. For the sake of simplicity to general readership, we do not wish to include this last part in the present study (it will, however, be visible to this response to reviewer file for the most curious readers). The rest is commented on in the main text (pages 6-7).

Reviewer-only Figure 1. U87 were transfected with siRNA control or targeting IRE1, XBP1 or PERK as indicated. 72 hours later, cells were treated with 100 ng/mL CD95L. % of cell death was defined as the % of Cytotox red-positive cells as detected by the Incucyte. Two independent experiments are shown (mean with range).

3. The specificity of MKC8866 at the concentration used (30 μ M) is unclear. What effect does MKCC have on sensitivity towards Fas-induced apoptosis, similar to the type of experimental set up presented in Figure 5A, 5B?

4. Similarly, what effects does MKC8866 (at 30 μ M) have on key Fas pathway determinants, such as Fas, FLIPL, FLIPs, Caspase-8, FADD, RIPK1, A20, CYLD, cIAP-1, cIAP-2 and Bid? There are many points at which MKC8866 could influence the outcome of Fas receptor engagement beyond the receptor itself.

Regarding the specificity of MKC8866, this drug has been optimized and refined from a family of IRE1-specific endoribonuclease inhibitors initially obtained from a chemical library screen (Volkman *et al*, 2011; Sheng *et al*, 2019, 2019). This salicylaldehyde analog has already shown to be effective in multiple cancer models including breast (Logue *et al*, 2018; Almanza *et al*, 2022), prostate (Sheng *et al*, 2019), pancreatic (Lv *et al*, 2023) cancers and it is currently used in clinical trial in breast cancers (NCT03950570), other tumors (NCT05154201, in combination with the standard of care) and IPF (NCT04643769). We have also recently demonstrated its efficacy in a GB mouse model (Le Reste *et al*, 2020). It is therefore a widely used IRE1 inhibitor, including in the dose range 10-30 μ M used in this study (e.g., (Logue *et al*, 2018; Almanza *et al*, 2022)). We therefore do not think it is in the scope of this manuscript to re-assess this compound's specificity.

Expanded view Figure 1C and D. U87 or SUM159 cells were pre-incubated for 1 hour with 1 μ g/mL of actinomycin D and further treated with 10 μ M MKC-8866 or 10 μ M Z4 for 1 hour followed by 2 hours treatment with 1 μ g/mL tunicamycin as indicated. CD95 mRNA expression level, normalized to GAPDH, was expressed as fold of value obtained for control (actinomycin-only treated samples). Mean \pm SEM, n=3-4. Unpaired t-test (for comparing TM and TM+MKC groups for C or TM and TM+Z4 groups for D), C * p=0.0461 for U87, *p=0.0437 for SUM159, D, * p=0.0241 for U87, (ns, p=0.0538 for SUM159).

To further evaluate the impact of pharmacologic inhibition of IRE1 on CD95 expression in cells, we have now tested an additional IRE1 inhibitor. This novel IRE1 kinase inhibitor has been developed in the laboratory <https://doi.org/10.1016/j.isci.2023.106687> and shown to blunt IRE1 activity in GB cells. We observe that Z4, like MKC, limits tunicamycin-induced decrease of CD95 expression (new **Figure EV1D**, see also **EV1C**) in U87 cells. This is not clearly observed in SUM159 cells, likely owing to the fact that SUM159 express higher level of IRE1 than U87 (data not shown) and that Z4 is less potent than MKC-8866 in inhibiting IRE1 activity (Pelizzari-Raymundo D., 2022).

Moreover, we have evaluated the impact of both MKC-8866 (30 μ M) and Z4 (25 μ M) towards basal CD95 protein expression (new **Figure 5C, EV5C**). The results highlight that both MKC-8866 and Z4 induce an accumulation of CD95 from 48 hours onwards, again pointing towards a dominance of the RIDD branch towards CD95 expression control. As requested, we have also analysed the expression of several other components of the CD95 signaling pathway (new **Appendix Figure S4B**). Note that we could not satisfactorily detect cIAP2, which has therefore not been included in the final analyses. In addition to the reviewer list, we have also included HOIP, as a key component of the LUBAC, another major E3 ligase regulating the signalling of several DR, including CD95 (Lafont *et al*, 2017; Taraborrelli *et al*, 2018; Tokunaga & Iwai, 2012). Interestingly, MKC-8866 reduction of both cFLIPL and cFLIPS expression, which might also contribute to modulation of apoptotic signalling induction. No other components among those tested were affected by IRE1 inhibition.

New Figure 5C (top left), new Expanded view Figure 5C (bottom left), new Appendix figure S4B (right). 5C. EV5C. U87 cells treated with DMSO, 30 μ M MKC-8866 (**5C, S4B**) or 25 μ M Z4 (**EV5C**) for the indicated times. Lysates were analysed by western blot with the indicated antibodies. One representative out of at least three independent experiments is shown.

As also suggested, we have evaluated the impact of MKC-8866 treatment on CD95L-induced cell death. As presented in new **Figure 5D**, MKC-8866 sensitized U87 cells to CD95L-induced cell death, in line with its ability to promote CD95 expression.

New Figure 5D. U87 cells treated with 30 μ M MKC-8866 or DMSO were further treated with 1 μ g/mL CD95L for the indicated times. Cell death was evaluated using Cytotox red positivity. Mean \pm SEM, n=3.

The results obtained with MKC-8866 and Z4 are described in pages 6-7 of the main text in the revised manuscript.

Minor issues:

1. For the Fas mRNA cleavage experiments presented in Figure 2, there are no irrelevant control mRNAs to allow the reader to judge whether the effects presented are specific to Fas mRNA or are commonly observed for many mRNAs at these amounts of IRE1 (1 μ g, 0.5 μ g, which appear high).

The expression of Fas mRNA was already normalized to GAPDH (which does not seem to vary upon incubation with IRE1). We nevertheless tested the expression levels of additional “irrelevant” RNAs in the initial samples as suggested by the reviewer. As shown below (new Appendix Figure S1), none of these were significantly affected by IRE1, implying that the effect observed on CD95 mRNA follows a specific trend which cannot be considered as a common one.

Appendix Figure S1- IRE1 does not cleave all RNAs *in vitro*. A. RNA (2 μ g) extracted from U87 cells was incubated with the indicated amounts of recombinant IRE1 for 1 hour. The indicated mRNAs were then quantified by RT-qPCR and normalized to GAPDH. Mean \pm SEM, n=3.

The amount of recombinant human IRE1 used in this experiment corresponds to 0.012pmol (1 μ g) and 0.0058pmol (0.5 μ g), for a total amount of 2 μ g total RNA. These concentrations were based on previous assays performed *in vitro* in our lab in which we have used 75 pmol of IRE1 to cleave 1 μ mol of IRE1 synthetic fluorescent substrate (as previously reported in the literature). The systematic determination of IRE1 RNase specific activity (rather than quantity) allowed us to maintain consistency throughout from batches to batches of commercial recombinant human IRE1 protein.

Reviewer #1 (Significance (Required)):

General assessment: this is an interesting study, as there is little knowledge currently concerning how the UPR influences Fas expression or Fas-dependent outcomes. However, the impact of this work is limited by the overexpression approaches used, which could produce artifactual results, as well as the contradictory message of the study.

Although we think that the message of the manuscript is indeed complex (since there is an opposite regulation of CD95 expression by RIDD and XBP1s), the work initially presented did not rely exclusively on overexpression approaches as our genetic-based results were also already comforted by the use of a pharmacologic inhibitor of IRE1 (**in cell models and *in vivo***). The revision experiments performed have now substantially strengthened our initial conclusions. Indeed, we now demonstrate this dual effect of IRE1 using siRNA-based approach and a second pharmacologic inhibitor of IRE1.

Advance: the advance reported here is relatively modest and limited in scope due to the inconclusive nature of the data presented.

Audience: this study will be of interests to specialists in the UPR and cell death communities.

We thank again the reviewer for acknowledging the overall novelty of our work and for suggesting experiments to improve this study. We also believe that this work might lead to further characterization of the impact of IRE1 inhibition in tumors. Since already an IRE1 inhibitor is tested in clinical trials, our work could constitute a founding stone for better stratifying the tumors and therefore likely enhance the efficacy of the treatment.

Reviewer #2 (Evidence, reproducibility and clarity (Required)):

The authors address here for the first time the connection between CD95, which is known as Fas, and ER stress. The role of another DR, TRAIL-R2 has been already reported, but this is the first study uncovering the link between Cd95 system and ER stress. The study is performed on the high level and supported by all necessary controls. They find the connection between IRE1 and CD95 and show that it might play a role in Cd95 signaling and attenuate CD95-mediated cell death. Further, the correlation between CD95 expression and IRE is found in tumors. Importantly the authors find out the connection between XBP1 and CD95 expression, which was not reported to date. Hence, it is a very important and highly essential piece of research. We thank the reviewer for these very positive comments and the acknowledging of the novelty and importance of our study.

However, I would like to clarify the several issues:

1: Figure 1. Tunicamycin obviously leads to deglycosylation of CD95, which is indicated by the appearance of 35 Kda band. This should be highlighted and commented.

We agree, we have now added the following sentence in the main text (page 4) of the revised version of the manuscript: "in accord with previous literature, tunicamycin treatment was also accompanied by a decrease of N-glycosylated CD95 (Shatnyeva *et al*, 2011)." This observation might be relevant in terms of signaling as previous work showed that expression of CD95 N118Q unglycosylated mutant led to normal FADD recruitment into DISC, but slightly reduced caspase 8 cleavage upon stimulation with an anti-Fas agonistic antibody (described in Shatnyeva *et al*, 2011).

2. Figure 2c, d. The piece of mRNA structure, which is synthesized, might have the different secondary structure and might be not cleaved by IRE, accordingly. More detailed comments have to be provided in this regard.

The model depicted in Figure 2B is a predicted computational secondary structure of CD95 mRNA. In the experiments performed in Figure 2A, C and D the mRNA was extracted from U87 cells prior to incubation with recombinant IRE1 and the resulting products analyzed using RT-qPCR with primers flanking different portions of the CD95 mRNA sequence. For Figures 2C and D, the primers used flank the two sites which were predicted to be cleaved by IRE1 based on previous work from our lab (Voutetakis, In preparation). One of them seems to be the predominant over the other. Even though we cannot exclude that additional sites can be targeted beyond this and that CD95 mRNA might adopt a different conformation in a cellular context, the fact that the amplification of CD95 sequence is reduced in samples pre-incubated with recombinant IRE1 strongly suggests that IRE1 is indeed able to cleave CD95 mRNA within

these regions *in vitro*. Whether this sole site is targeted in cells should constitute an object of a future study. We have modified the main text to further explain this point (page 4-5).

3. Figure 3. Caspase-8-3 western blots show beautiful effects but did authors see some effects further downstream, eg on PARP1 cleavage? Was cell death (not viability) measured as well? Can you comment on this?

This is absolutely right; we have now included the cleavage of PARP-1 in this setting as suggested. The results obtained confirm the fact that caspase activation is enhanced in IRE1 DN-overexpressing cells as compared to WT (PARP-1 staining now included in updated **Figure 3C**). Interestingly, this data also hints to a role for IRE1 in controlling PARP-1 expression, which if confirmed in additional models, might be interesting considering the known relationship between the UPR and the response to DNA damage response (González-Quiroz *et al*, 2020).

Updated Figure 3C. U87 WT or expressing IRE1DN were treated with 250 ng/mL CD95L for the indicated times. Lysates were analysed by western blot. One experiment representative of three independent ones is shown.

Given the morphology of the cells observed in the viability experiments, we expected a similar trend using cell death assays. However, we do agree with the reviewer that this needed be proven experimentally, so we have performed the experiments in Figure 3 again using cell death assays as a read out. Please note that we have decided to mainly show cell death experiments up to 12 hours in this study (as opposed to mainly 24 hours for viability) as in most cases this is the time at which the maximum cell death is reached. Beyond that, for most experiments, a decrease of fluorescence (due to full necrosis of the cells and diffusion of small cellular debris) is detected by the Incucyte beyond this point (see **Reviewer-only Figure 2**)

Reviewer-only Figure 2 (corresponds to the full kinetic from new Figure 3B). U87 WT or expressing IRE1DN were treated with 1 mg/mL CD95L for 12 hours. % of cell death was defined as the % of Cytotox red-positive cells as detected by the Incucyte. Mean \pm SEM of 3 independent experiments.

The results obtained show that the expression of dominant negative forms of IRE1 (IRE1DN or IRE1Q780*) sensitize the different cell lines used to CD95L-induced cell death (**new Figures 3B and D**). Note that for RADH87 cells (**new Figure EV2B**), the concentration for birinapant was increased to 100 nM final concentration (and not 50 nM as used in the initial viability experiment, now **Appendix Figure S3C**), since 100 nM did not yield any significant increase in

cell death induction. We suspect that the loss of viability observed in the initial experiments using 100 nM of birinapant was partly due to a cytostatic effect. These data is now discussed on page 5.

New Figures 3B (left), 3D (middle) and EV2B (right). **3B.** U87 WT or expressing IRE1DN were treated with 1 mg/mL CD95L for 12 hours. % of cell death was defined as the % of Cytotox red-positive cells as detected by the Incucyte. Mean \pm SEM of 3 independent experiments. **3D.** RADH85 control (EV), stably expressing IRE1Q780* or IRE1WT were treated with 500 ng/mL CD95L for 12 hours. % of cell death was defined as the % of Cytotox red-positive cells as detected by the Incucyte. Mean \pm SEM of 3 independent experiments. **EV2B.** RADH87 control (EV), stably expressing IRE1Q780* or IRE1WT were pre-treated with 200 nM (2X) of Birinapant for 1 hour prior to addition of 1 mg/mL CD95L for 24 hours. % of cell death was defined as the % of Cytotox red-positive cells as detected by the Incucyte. Mean \pm SEM of 3 independent experiments.

4. Did the authors looked at the DISC assembly? Did they detect some differences there? We thank the reviewer for this suggestion. We have now evaluated the DISC formation, by performing immunoprecipitation of CD95 upon stimulation with CD95L. The results obtained are in accord with our initial observations and demonstrate that cells overexpressing a dominant negative form of IRE1 (IRE1Q*), as compared to empty vector or IRE1WT-overexpressing cells, display an increased DISC formation (**new Figure 3E**).

Figure 3E. Empty vector (EV), IRE1WT- or IRE1Q780*-expressing RADH85 were treated with 500 ng/mL CD95L for the indicated times. The DISC immunoprecipitated using an anti-CD95 antibody prior to western blot analysis. One experiment representative of two, independent ones. *indicates an unspecific band.

Reviewer #2 (Significance (Required)):

This is an excellent study. The authors address here for the first time the connection between CD95, which is known as Fas, and ER stress. The role of another DR, TRAIL-R2 has been already reported, but this is the first study uncovering the link between Cd95 system and ER stress. The study is performed on the high level and supported by all necessary controls. This is an important advance for the death receptor field.

We thank this reviewer again for these very positive comments and insightful suggestions to improve our work.

References

- Almanza A, Mnich K, Blomme A, Robinson CM, Rodriguez-Blanco G, Kierszniowska S, McGrath EP, Le Gallo M, Pilalis E, Swinnen JV, *et al* (2022) Regulated IRE1 α -dependent decay (RIDD)-mediated reprogramming of lipid metabolism in cancer. *Nat Commun* 13: 2493
- Chang TK, Lawrence DA, Lu M, Tan J, Harnoss JM, Marsters SA, Liu P, Sandoval W, Martin SE & Ashkenazi A (2018) Coordination between Two Branches of the Unfolded Protein Response Determines Apoptotic Cell Fate. *Mol Cell* 71: 629-636.e5
- González-Quiroz M, Blondel A, Sagredo A, Hetz C, Chevet E & Pedoux R (2020) When Endoplasmic Reticulum Proteostasis Meets the DNA Damage Response. *Trends Cell Biol* 30: 881–891
- Lafont E (2020) Stress Management: Death Receptor Signalling and Cross-Talks with the Unfolded Protein Response in Cancer. *Cancers* 12: 1113
- Lafont E, Kantari-Mimoun C, Draber P, De Miguel D, Hartwig T, Reichert M, Kupka S, Shimizu Y, Taraborrelli L, Spit M, *et al* (2017) The linear ubiquitin chain assembly complex regulates TRAIL-induced gene activation and cell death. *EMBO J* 36: 1147–1166
- Le Reste PJ, Pineau R, Voutetakis K, Samal J, Jégou G, Lhomond S, Gorman AM, Samali A, Patterson JB, Zeng Q, *et al* (2020) Local intracerebral Inhibition of IRE1 by MKC8866 sensitizes glioblastoma to irradiation/chemotherapy in vivo. 841296
- Logue SE, McGrath EP, Cleary P, Greene S, Mnich K, Almanza A, Chevet E, Dwyer RM, Oommen A, Legembre P, *et al* (2018) Inhibition of IRE1 RNase activity modulates the tumor cell secretome and enhances response to chemotherapy. *Nat Commun* 9: 3267
- Lv X, Lu X, Cao J, Luo Q, Ding Y, Peng F, Pataer A, Lu D, Han D, Malmberg E, *et al* (2023) Modulation of the proteostasis network promotes tumor resistance to oncogenic KRAS inhibitors. *Science* 381: eabn4180
- Pelizzari-Raymundo D. DD (2022) A novel blood brain barrier-permeable IRE1 kinase inhibitor sensitizes glioblastoma to chemotherapy in mice. *ChemRxiv* <https://chemrxiv.org/engage/chemrxiv/article-details/61d97d56db142ef572b84e87>
- Shatnyeva OM, Kubarenko AV, Weber CEM, Pappa A, Schwartz-Albiez R, Weber ANR, Krammer PH & Lavrik IN (2011) Modulation of the CD95-Induced Apoptosis: The Role of CD95 N-Glycosylation. *PLoS One* 6: e19927
- Sheng X, Nenseth HZ, Qu S, Kuzu OF, Frahnaw T, Simon L, Greene S, Zeng Q, Fazli L, Rennie PS, *et al* (2019) IRE1 α -XBP1s pathway promotes prostate cancer by activating c-MYC signaling. *Nat Commun* 10: 323
- Taraborrelli L, Peltzer N, Montinaro A, Kupka S, Rieser E, Hartwig T, Sarr A, Darding M, Draber P, Haas TL, *et al* (2018) LUBAC prevents lethal dermatitis by inhibiting cell death induced by TNF, TRAIL and CD95L. *Nat Commun* 9: 3910
- Tokunaga F & Iwai K (2012) LUBAC, a novel ubiquitin ligase for linear ubiquitination, is crucial for inflammation and immune responses. *Microbes Infect* 14: 563–72

Tsuru A, Imai Y, Saito M & Kohno K (2016) Novel mechanism of enhancing IRE1 α -XBP1 signalling via the PERK-ATF4 pathway. *Sci Rep* 6: 24217

Volkman K, Lucas JL, Vuga D, Wang X, Brumm D, Stiles C, Kriebel D, Der-Sarkissian A, Krishnan K, Schweitzer C, *et al* (2011) Potent and selective inhibitors of the inositol-requiring enzyme 1 endoribonuclease. *J Biol Chem* 286: 12743–55

Voutetakis K; D (In preparation) RNA sequence motif and structure in IRE1-mediated cleavage.

Dear Dr. Lafont,

Thank you for your patience over the holiday period. The revised manuscript was evaluated by two subject experts who based their evaluation on the previous referee reports and your revision (their short reports are included below).

We are very pleased indeed to publish your paper following minor additional (textual) revision:

We would encourage you to add to and to revise the discussion section according to the suggestions of arbitrating referee 3. Please note also that the referee makes a number of further reaching suggestion: 'it will be interesting to measure DR5 in the in vivo and in vitro experimental system when IRE1 is manipulated, in addition to analyze possible promoter regions of the CD95 promoter that may be regulated by XBP1. Performing a bioinformatic analysis across species will be easy.' These three points can be usefully noted textually in the discussion, or added as data if such data/bioinformatic analysis is available in the meantime.

Please note that in figure 4, figure legend for figure 4c is incorrectly labeled as 4d.

I look forward to seeing a new revised version of your manuscript as soon as possible.

We will pass to production the final paper as soon as we received your revisions. Production typically takes less than two weeks.

best wishes,

Bernd Pulverer

~~~~~  
Bernd Pulverer, Ph.D.  
Chief Editor, EMBO Reports  
EMBO  
Meyerhofstrasse 1, D-69117 Heidelberg  
Tel: +4962218891501  
bernd.pulverer@embo.org  
~~~~~

Arbitrating Referee #3:

I was requested to evaluate the comments by the reviewers and responses from the authors.

I believe that most comments from reviewer 1 were addressed by the authors. I agree with most critics from reviewer 1, specifically in the need to perform complementary experiments using an alternative approaches to modify IRE1 signaling that do not rely on the over-expression of mutant proteins (i.e. siRNAs), in addition to the overexpression of XBP1s, and the use of pharmacological ER stressors. These requests were included now in the revised version. Now the study is complemented with alternative experiments.

Since IRE1 has dual roles on CD95 expression through RIDD (mRNA degradation) and XBP1s (gene upregulation), it is expected to obtain different outcomes in different experimental systems. Data is data. This is also known for other experimental system, including insulin production where RIDD regulates its mRNA levels, XBP1 induces its expression, and in addition controls the pathways involved in insulin maturation and signaling. This could be included in the discussion. These type of example showing different layers of regulation suggest that there is an evolutionary conserved pathway that has relevant physiological roles.

Regarding reviewer two, it was more supportive of the study and requested very precise experiments to improve the paper. Among them, the most difficult ones were related to measure DISC assembly, which were properly performed.

For my criteria, the study meets standards for publication. I am not sure how general the findings are for EMBO Reports. I believe this should be addressed by the editor. But importantly there is little information about the role of death receptors in ER stress-mediated apoptosis and its relation to the UPR. This is why this study will add value to the field since most studies are focused on DR5. It is clear that ER stress-mediated cell death is highly complex and mediated by multiple pathways that together contribute to cell death.

I was not requested to me to generate additional questions. It will be interesting to measure DR5 in the in vivo and in vitro experimental system when IRE1 is manipulated, in addition to analyze possible promoter regions of the CD95 promoter that may be regulated by XBP1. Performing a bioinformatic analysis across species will be easy.

I suggest improving the discussion to compare with other systems may help putting the study on broader context. The discussion is very poor. Please improve the discussion to mention studies on DR5 and other links between death receptors and ER stress.

Arbitrating Referee #4:

The authors have comprehensively addressed the concerns raised and have included new data that have greatly improved the study. I recommend the paper for publication.

EMBO Reports**Manuscript number:** EMBOR-2023-57277V2*In black, the reviewer comments.**In blue, the authors' responses.***Arbitrating Referee #3:**

I was requested to evaluate the comments by the reviewers and responses from the authors. I believe that most comments from reviewer 1 were addressed by the authors. I agree with most critics from reviewer 1, specifically in the need to perform complementary experiments using an alternative approaches to modify IRE1 signaling that do not rely on the over-expression of mutant proteins (i.e. siRNAs), in addition to the overexpression of XBP1s, and the use of pharmacological ER stressors. These requests were included now in the revised version. Now the study is complemented with alternative experiments.

Since IRE1 has dual roles on CD95 expression through RIDD (mRNA degradation) and XBP1s (gene upregulation), it is expected to obtain different outcomes in different experimental systems. Data is data. This is also known for other experimental system, including insulin production where RIDD regulates its mRNA levels, XBP1 induces its expression, and in addition controls the pathways involved in insulin maturation and signaling. This could be included in the discussion. These type of example showing different layers of regulation suggest that there is an evolutionary conserved pathway that has relevant physiological roles. Regarding reviewer two, it was more supportive of the study and requested very precise experiments to improve the paper. Among them, the most difficult ones were related to measure DISC assembly, which were properly performed.

For my criteria, the study meets standards for publication. I am not sure how general the findings are for EMBO Reports. I believe this should be addressed by the editor. But importantly there is little information about the role of death receptors in ER stress-mediated apoptosis and its relation to the UPR. This is why this study will add value to the field since most studies are focused on DR5. It is clear that ER stress-mediated cell death is highly complex and mediated by multiple pathways that together contribute to cell death.

I was not requested to me to generate additional questions. It will be interesting to measure DR5 in the in vivo and in vitro experimental system when IRE1 is manipulated, in addition to analyze possible promoter regions of the CD95 promoter that may be regulated by XBP1. Performing a bioinformatic analysis across species will be easy.

I suggest improving the discussion to compare with other systems may help putting the study on broader context. The discussion is very poor. Please improve the discussion to mention studies on DR5 and other links between death receptors and ER stress.

We thank arbitrating referee #3 for their assessment. We have now addressed their comments in full, as further detailed below.

As advised, we have evaluated the presence of putative XBP1s-binding sites on murine and human FAS promoter regions (focusing on the 1500 bp upstream of the start codon) using the TFbind bioinformatic tool. This analysis revealed that both murine and human FAS promoter regions display several putative XBP1s-binding sites, even though the extent of conservation for each individual site is limited. This analysis is now depicted in Appendix Figures S5 and Appendix Table S1 and commented on as follows on page 6 of the main text: "Of note, the

promoter regions of both murine and human *FAS* displayed several potential XBP1s-binding sites, whilst each individual site is solely partially conserved between these two species (**Appendix Fig S5 and Table S1**).”

As advised by this reviewer, we have also updated the discussion to include more considerations on the link between TRAIL-R and UPR and to further develop the idea that dual regulation of gene expression by IRE1 could be functionally relevant in other cellular contexts (including insulin signalling), on pages 7 to 9 of the revised manuscript.

We thank arbitrating referee #3 again for their insightful suggestions.

Arbitrating Referee #4:

The authors have comprehensively addressed the concerns raised and have included new data that have greatly improved the study. I recommend the paper for publication.

We thank arbitrating referee #4 for their comments.

Dr. Elodie Lafont
Inserm
France

Dear Dr. Lafont,

I am very pleased to accept your manuscript for publication in the next available issue of EMBO reports. Thank you for your contribution to our journal.

When you receive the proofs, I would recommend two minor textual changes:

- 1) please remove 'pharmacological actionable' for the last line of the abstract, as it does not add to the information provided by this study
- 2) in the next text added you state 'solely partially conserved ' > please change to 'only partially conserved'.

Yours sincerely,

Bernd Pulverer

~~~~~  
 Bernd Pulverer, Ph.D.  
 Chief Editor, EMBO Reports  
 EMBO  
 Meyerhofstrasse 1, D-69117 Heidelberg  
 Tel: +4962218891501  
[bernd.pulverer@embo.org](mailto:bernd.pulverer@embo.org)  
 ~~~~~

Rev_Com_number: RC-2022-01800
 New_manu_number: EMBOR-2023-57277V3
 Corr_author: Lafont
 Title: IRE1 RNase controls CD95-mediated cell death